# Nur77 protects the bladder urothelium from intracellular bacterial infection

Christina A. Collins[1], Chevaughn Waller[2], Ekaterina Batourina[2], Lokesh Kumar[1], Cathy L. Mendelsohn ◉[2] & Nicole M. Gilbert ◉[1,3] ✉

Intracellular infections by Gram-negative bacteria are a significant global health threat. The nuclear receptor Nur77 (also called TR3, NGFI-B, or NR4A1) was recently shown to sense cytosolic bacterial lipopolysaccharide (LPS). However, the potential role for Nur77 in controlling intracellular bacterial infection has not been examined. Here we show that Nur77 protects against intracellular infection in the bladder by uropathogenic *Escherichia coli* (UPEC), the leading cause of urinary tract infections (UTI). Nur77 deficiency in mice promotes the formation of UPEC intracellular bacterial communities (IBCs) in the cells lining the bladder lumen, leading to persistent infection in bladder tissue. Conversely, treatment with a small-molecule Nur77 agonist, cytosporone B, inhibits invasion and enhances the expulsion of UPEC from human urothelial cells in vitro, and significantly reduces UPEC IBC formation and bladder infection in mice. Our findings reveal a new role for Nur77 in control of bacterial infection and suggest that pharmacologic agonism of Nur77 function may represent a promising antibiotic-sparing therapeutic approach for UTI.

The urinary tract is the second most common site of bacterial infection in humans, and uropathogenic *Escherichia coli* (UPEC) are the leading cause of UTI globally[1]. The high incidence of UTI results in sizeable health care costs, annually reaching billions of dollars globally and $2 billion in the USA alone[2]. Antibiotics remain the primary treatment option for UTI, precipitating the worldwide spread of antibiotic-resistant uropathogens. Even when initial treatment is effective, UTI remain highly recurrent. Nearly one in four women with an initial UTI will suffer recurrent UTI (rUTI) within 6 months, and up to 70% will experience rUTI within 1 year; half will have multiple recurrences over their lifetime[1,3], and the yearly risk of recurrence in children is 19–22%[4]. Adult and pediatric patients with a history of rUTI are often given prophylactic antibiotics to ward off future infection, which predictably leads to infection with antibiotic-resistant pathogens[5–7]. New therapies targeting host factors that promote resistance to UTI without exacerbating the rise of antibiotic-resistant organisms would be highly useful.

During cystitis, several uropathogens, including UPEC, can invade and survive within uroepithelial cells where the bacteria can avoid antibiotic exposure[8–10]. Urothelial cells use several mechanisms to eliminate invading bacteria, including activating programmed cell death-mediated exfoliation[11–14], inducing autophagy[15], or by expulsion of internalized organisms[16–18]. When these measures fail, UPEC can escape into the cytosol and replicate to form large intracellular bacterial communities (IBCs), from which a subset of organisms eventually flux out of the cell to perpetuate the infection[19,20]. Intracellular bacteria have been detected within shed urothelial cells during UTI in adults[21] and children[22]. Smaller collections of intracellular UPEC can persist in urothelial cells for months following resolution of bacteriuria[23], and these so-called quiescent intracellular reservoirs (QIRs) are considered a likely source of rUTI. In mice, uropathogen emergence can be triggered by bladder exposures to other urogenital bacteria[24] or upon catheterization[25]. In humans, up to two-third of sequential UTI episodes are caused by the same uropathogen strain, which is consistent with reservoir re-emergence. Bacteria have been detected in bladder biopsies from patients with a history of UTI, even after antibiotic treatment and

[1]Department of Pediatrics, Division of Infectious Diseases, Washington University School of Medicine, St. Louis, MO, USA. [2]Department of Urology, Columbia University Irving Medical Center, New York, NY, USA. [3]Center for Women's Infectious Disease Research, Washington University School of Medicine, St. Louis, MO, USA. ✉e-mail: gilbert@wustl.edu

confirmed clearance of bacteriuria[9,26]. Ideally, future UTI therapies would target UPEC intracellular infection.

The nuclear receptor Nur77, also known as NR4A1, TR3, or NGFI-B, is a transcription factor with important roles in inflammation, apoptosis, and cell proliferation[27–30]. Nur77 has been extensively studied as an key regulator of immune homeostasis and as an influencer of the balance between cell survival and death in immune cells[31,32] and epithelial cells[33,34]. The endogenous ligand for Nur77 was unknown until very recently, when it was discovered that Nur77 is a cytosolic sensor for intracellular LPS[35]. This suggests that Nur77 could participate in host responses to intracellular bacterial infections. However, despite a known role as a regulator of inflammation, Nur77 has almost exclusively been examined in the context of noninfectious diseases such as cancer, leaving the importance of Nur77 in bacterial infections largely unknown. Prior to the discovery of Nur77-LPS binding, we reported that Nur77 expression was induced in the bladder in a mouse model of recurrent urinary tract infection (UTI) arising from intracellular reservoirs of uropathogenic *Escherichia coli* (UPEC)[36]. Furthermore, we observed differences in the rate of rUTI between WT mice and mice with germline deficiency of Nur77 (Nur77-KO). Mice lacking Nur77 maintained higher levels of UPEC intracellular reservoir infection following exfoliation-inducing exposures that successfully reduced UPEC reservoir titers in WT mice[36]. These results suggested that Nur77 expression in the bladder may modulate intracellular UPEC infection.

In this study, we canvassed available single-cell RNAseq datasets to find that Nur77 is ubiquitously expressed in multiple cell types in the bladders of mice and humans. Building on our prior work, we demonstrate in a preclinical mouse model that Nur77 is necessary to limit UPEC bladder infection. Pharmacological treatment with a Nur77 agonist in human urothelial cells in vitro blocked endocytosis and prevented UPEC intracellular invasion and limited UPEC UTI in vivo in the mouse model of UTI. These complimentary findings provide evidence that Nur77 is important for controlling UPEC infection in the bladder and may represent a novel therapeutic target for UTI.

## Results

### Results from knockout mice implicate Nur77 in persistent UPEC bladder infection

Given the importance of Nur77 in multiple cellular processes that are known to be involved in the UPEC UTI pathogenic cascade, and the results from our prior rUTI model[36], we reasoned that Nur77 might modulate outcomes of UPEC UTI. Therefore, we performed experimental UPEC UTI in mice with germline deficiency of Nur77 (Nur77-KO). First, we confirmed that there were no baseline differences in gross urothelial morphology that could affect UTI outcomes. There were no apparent alterations in urothelial architecture at baseline in the absence of Nur77. Both WT and Nur77-KO bladders exhibited typical staining of uroplakin 2 (urothelial cells), p63 (intermediate and basal cells), and keratin 5 (basal cells) (Fig. S1). Female 6–7-week-old wild-type (WT) C57BL/6 or Nur77-KO mice were transurethrally inoculated with UPEC strain UTI89 (Fig. 1A), and UPEC colony-forming units (CFU) were enumerated in urine and bladder tissue. Of note, C57BL/6 mice typically resolve UPEC bacteriuria over the course of one to several weeks; meanwhile, UPEC that have successfully invaded the urothelium and established QIRs are maintained in the bladder for months while urine cultures remain negative[23,24]. Here, Nur77-KO mice exhibited approximately tenfold lower UPEC bacteriuria 24 hpi, but this difference had resolved by 1 wpi (Fig. 1B). There was no difference in the duration of bacteriuria, with most WT and Nur77-KO mice clearing UPEC from urine by 3 wpi (Fig. 1C). The difference in UPEC bacteriuria 24 hpi was not reflected in bacterial loads in bladder tissue, which were equal 24 hpi and 1 wpi between WT and Nur77-KO mice (Fig. 1D). In contrast to these results at acute time points, when mice were examined later (3 and 4 wpi; after resolution of UPEC bacteriuria), Nur77-KO mice harbored threefold more UPEC CFU compared to WT

controls ($P < 0.05$ at 3 wpi, $P < 0.01$ at 4 wpi, Fig. 1E). These data suggest that Nur77 acts to limit UPEC infection and persistence in bladder tissue.

### Cellular expression of Nur77 in the mouse and human bladder
Nur77 is broadly expressed by multiple cell types. Our previous bulk RNA sequencing (RNAseq) analysis reported Nur77 expression in bladders from naive and UPEC-infected mice and in mice transurethrally inoculated with *Gardnerella* (Fig. S2A). To gain further insight into which cell types in the bladder could be mediating the effect of Nur77 on UPEC titers, we examined published and publicly available single-cell transcriptomics datasets. *Tabula Muris* contains single-cell (sc)RNAseq transcript analysis of 20 tissues and organs from C57BL/6 mice, including the bladder[37]. Nur77 expression was evident in cells from all *Tabula Muris* cell clusters in the bladder (Fig. S2B). Notably, among the 100,000 cells analyzed from all tissues in *Tabula Muris*, the "bladder cell" clusters had the second-highest expression of Nur77. Yu et al. performed high-throughput, droplet-based scRNAseq analysis of healthy C57BL/6 mouse and human bladder tissue[38]. Their analysis classified 15 distinct cell clusters in the mouse bladder and 16 clusters in the human bladder. Nur77 expression was detected in leukocytes (monocytes, dendritic cells, T cells, and B cells), all three layers of the urothelium (basal, intermediate, umbrella), and other bladder cell types including fibroblasts, smooth muscle cells, and endothelial cells (Fig. S2C).

### Absence of Nur77 does not alter bladder immune cell populations before or during UTI
Nur77 is an important regulator of host inflammatory responses[27,29,32,39]. Therefore, we reasoned that the increase in UPEC bladder burden in Nur77-KO mice 3 and 4 wpi could be due to a deficiency in inflammation. First, we established that there were no baseline differences in the populations of myeloid or lymphoid cells in the bladders between naive WT and Nur77-KO mice (Fig. S3). We examined bladders from WT and Nur77-KO mice 24 hpi and 4 wpi by histology and flow cytometry. Both WT and Nur77-KO mice displayed the expected histological inflammation and edema 24 hpi (Fig. 2A). There were no significant differences in CD45+ cell numbers and only a slight, yet statistically significant, decrease in the proportion of B cells in Nur77-KO mice compared to WT (Fig. 2C). We also measured UPEC uptake by immune cells as previously described[40] (Fig. 2D), but there was no difference in the distribution of effector cells that were UPEC+ (Fig. 2E) or in the percentage of each myeloid cell type that were UPEC+ (Fig. 2F). These data suggest that Nur77 is not necessary for immune cell expansion in the bladder or phagocytosis of UPEC during the acute stages of infection. Following resolution of bacteriuria, histological inflammation and edema resolved in both WT and Nur77-KO mice (Fig. 2G), and the number of CD45+ cells in the bladder returned to baseline levels 4 wpi (Fig. 2H). There were no significant differences in the numbers or proportions of CD45+ cell types 4 wpi (Fig. 2I) when comparing infected WT and Nur77-KO mice. However, differences became apparent when comparing naive to infected mice in each genotype. WT-infected mice had an increase only in the proportion of monocytes 4 wpi (Fig. 2J, K), whereas Nur77-KO-infected mice had significantly increased PMNs, DCs, monocytes, and macrophages in the bladder 4 wpi (Fig. 2L). There was a trend toward increased NK cells 4 wpi in WT mice (Fig. 2K), and this difference was statistically significant in Nur77-KO mice, with a concomitant decrease in T cells (Fig. 2M). The observation that Nur77-KO mice had proportionally increased myeloid cell populations in the bladder is consistent with the heightened UPEC bacterial burden in Nur77-KO bladders at this time point.

### UPEC IBC formation is increased in the absence of Nur77
The establishment of the persistent reservoirs of UPEC in the bladder that are present 4 wpi requires invasion of urothelial cells during the

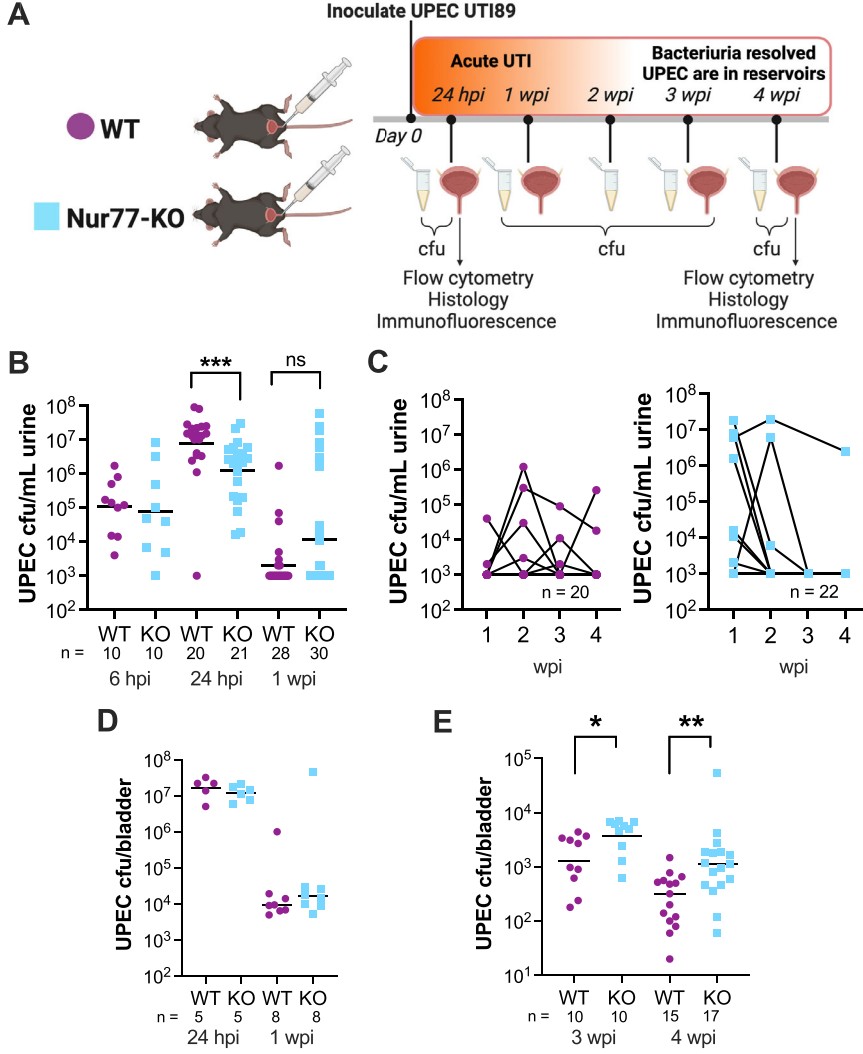

**Fig. 1 | Mice lacking Nur77 permit increased UPEC persistence in bladder tissue.**
**A** Schematic of UPEC UTI mouse model. **A** Created with BioRender.com released under a Creative Commons Attribution-NonCommercial-NoDerivs 4.0 International license. **B** UPEC titers in urine during the first week of infection; data combined from five independent experiments. Kruskal–Wallis $P < 0.0001$; ***$P = 0.0010$ by two-tailed Mann–Whitney $U$ test. **C** UPEC titers in urine collected weekly; data combined from two independent experiments. **D** Acute infection UPEC titers in bladder tissue homogenates; data from one experiment per time point. **E** Quiescent reservoir UPEC titers in bladder tissue homogenates; data from two independent experiments per time point. Central tendency lines in graphs (**B**, **D**, **E**) denote geometric mean values. Kruskal–Wallis $P < 0.0001$; *$P = 0.0119$, **$P = 0.0035$, by two-tailed Mann–Whitney $U$ test. Source data are provided as a Source Data file.

acute phase of the infection. We hypothesized that the heightened UPEC burden in the bladders of Nur77-KO mice 4 wpi could result from enhanced invasion earlier during infection. To address this hypothesis, we first performed ex vivo gentamicin protection assays that separately enumerated intracellular and extracellular UPEC CFU. Compared to WT mice 3 hpi, Nur77-KO mice had significantly lower UPEC CFU in both the extracellular (washes) and intracellular (gentamicin-protected) bladder compartments. But, by 6 and 24 hpi, there were no differences in either extracellular or intracellular UPEC CFU (Fig. S4). In WT mice, intracellular UPEC at 6 and 24 hpi are known to be primarily present in intracellular bacterial communities (IBCs), that are formed by intracellular replication following invasion, each of which contain ~$10^5$ bacterial cells. Therefore, CFU data from ex vivo gentamicin protection assays cannot distinguish between UPEC invasion and subsequent intracellular replication. Since each IBC in the bladder represents a successful UPEC invasion event, to examine invasion more specifically, we enumerated IBCs 6 hpi and 24 hpi in LacZ-stained splayed bladders. Despite having lower UPEC CFU 3 hpi in gentamicin protection assays, we observed an approximately twofold increase in

the number of IBCs in Nur77-KO mice compared to WT at both 6 and 24 hpi (Figs. 3A and S5). Histological and immunofluorescence microscopy analyses confirmed the presence of IBCs in superficial urothelial cells of Nur77-KO mice with similar morphology as seen in WT mice (Fig. 3B). These results suggest that Nur77 may function in urothelial cells early during infection to limit the UPEC urothelial cell invasion that is needed for IBC formation and eventually leads to persistent intracellular reservoirs.

## A small-molecule Nur77 agonist inhibits UPEC intracellular urothelial infection in vitro

The increase in IBC numbers in Nur77-KO mice suggested that Nur77 could function in urothelial cells to limit intracellular UPEC burden. To test this idea, we performed infection experiments in cultured 5637 urothelial cells. First, we performed siRNA knockdown of Nur77 and compared UPEC intracellular infection to control siRNA cells. Despite achieving significant reduction in Nur77 transcript, there were no significant differences in UPEC intracellular infection in these experiments, suggesting that Nur77 is not essential to permit invasion and

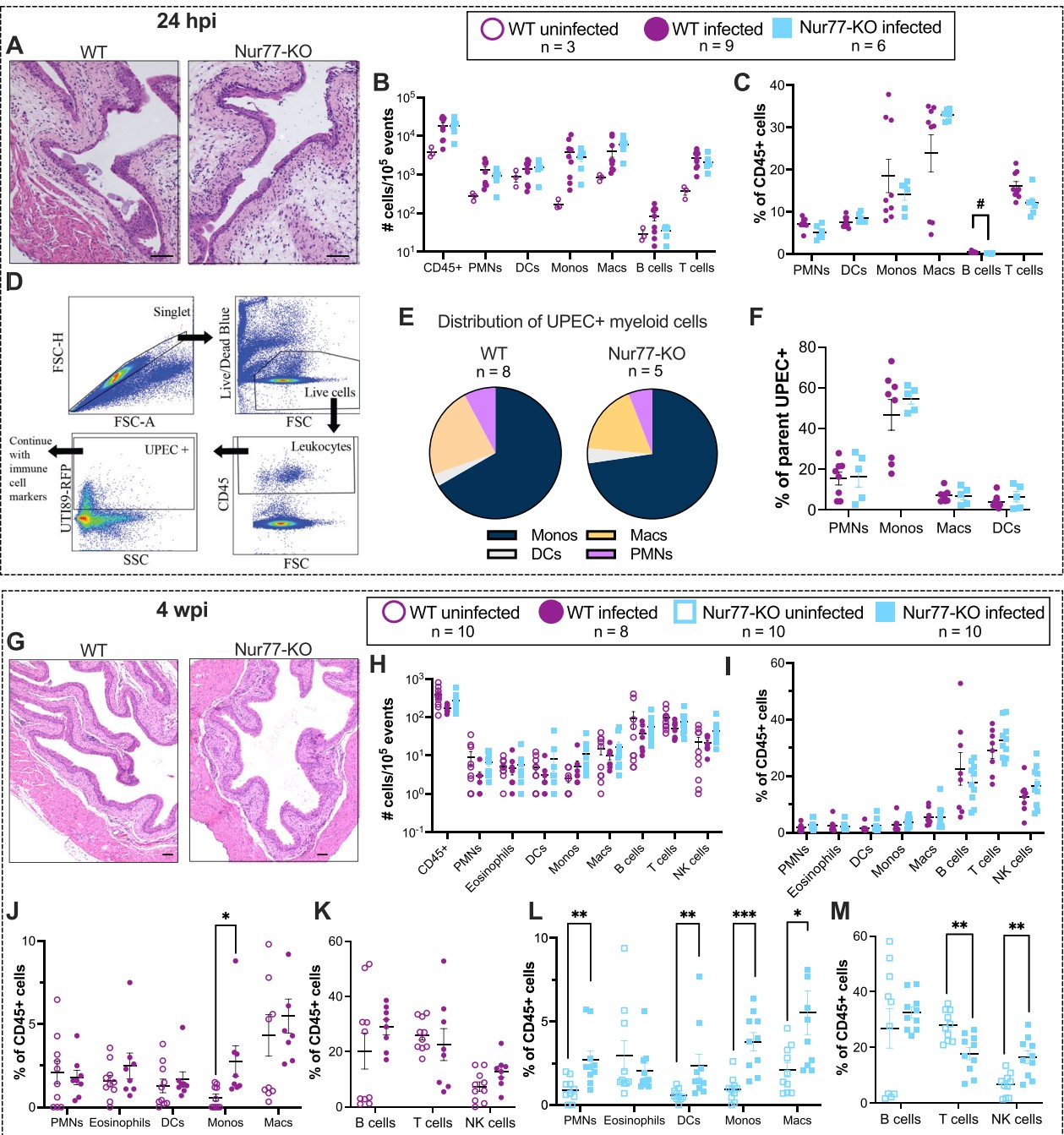

**Fig. 2 | WT and Nur77-KO mice display similar patterns of immune cell populations in the bladder during and following the resolution of UPEC UTI.**
**A**–**F** Bladders collected 24 h after mock or UTI89 infection. **A** H&E staining of formalin-fixed paraffin-embedded bladder tissue sections. Scale bar = 100 μm. **B**, **C** Comparison of the numbers (**B**) and relative abundances (**C**) of immune cell types between WT and Nur77-KO mice, combined from two independent experiments. **D** Gating strategy for UTI89-RFP+ immune cells. **E** Distribution of UPEC+ myeloid cells in WT and Nur77-KO bladders. **F** Percentage of each myeloid cell type that was UPEC + . **G**–**M** Bladders collected 4 weeks after mock or UTI89 infection. All infected mice in this analysis resolved bacteriuria and harbored quiescent UPEC

reservoirs at this time point; data combined from two independent experiments. **G** H&E staining of formalin-fixed paraffin bladder tissue sections. Scale bar = 100 μm. **H, I** Comparison of the numbers (**H**) and relative abundances (**I**) of immune cell types between WT and Nur77-KO mice. **J**–**M** Comparison of myeloid (**J**, **L**) and lymphoid (**K**, **M**) relative abundances between uninfected and infected mice of each genotype. WT (**J**, **K**); Nur77-KO (**L**, **M**). All graphs show mean +/− SEM. **J** *P = 0.005256; **L** ***P = 0.00119, **P = 0.000985 (PMNs), P = 0.000725 (DCs), *P = 0.010793; (**M**) **P = 0.003107 (T cells), **P = 0.002273 (NKs) by two-tailed Mann–Whitney U test with Holm–Šídák method. Source data are provided as a Source Data file.

intracellular infection in vitro (Fig. S6). Next, we used the Nur77 agonist cytosporone B (CsnB)[41] to activate Nur77. CsnB, is an octaketide isolated from an endophytic fungus that was the first identified natural ligand for Nur77[41]. CsnB has a strong affinity for Nur77 and has been widely used to specifically activate Nur77 in vitro[41–46]. Therefore, CsnB would allow us to determine the effect of activating Nur77 at various

time points during UPEC infection. Given that Nur77-KO mice had increased UPEC infection, we hypothesized that activation of Nur77 with CsnB would decrease UPEC urothelial infection. Before performing infection experiments, we confirmed that incubation with CsnB did not affect UPEC viability (Fig. S7A) or the bacterial hemagglutination (HA) titers that are a measure of UPEC type 1 pili expression

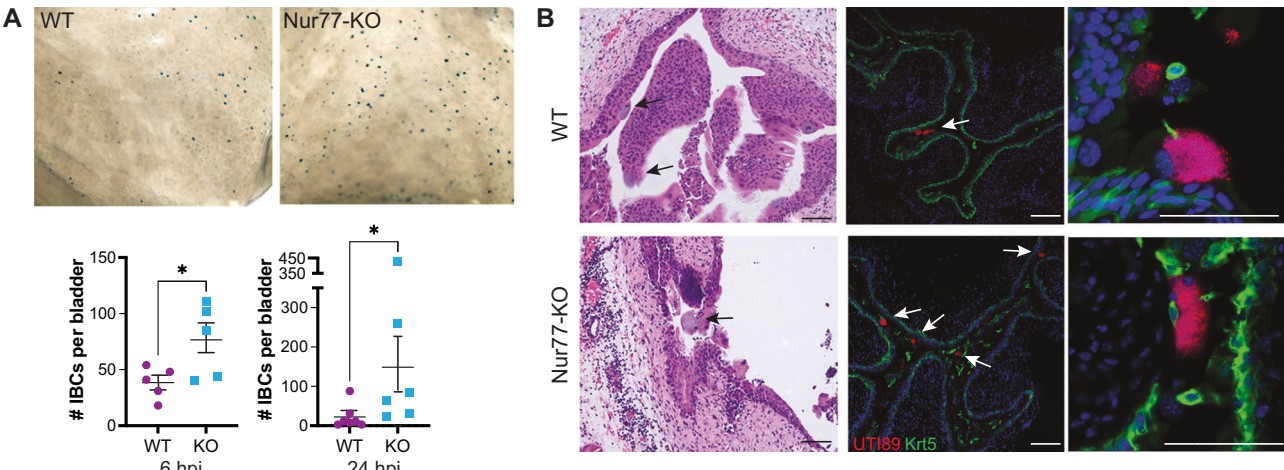

**Fig. 3 | UPEC produce more intracellular bacterial communities (IBCs) in the bladders of Nur77-KO compared to WT mice. A** Representative images of splayed bladders collected 6 hpi and stained with X-gal to visualize IBCs. IBCs were enumerated in three independent experiments (see Fig. S5 for data from third experiment). Graphs show the mean +/− SEM. 6 hpi WT and KO $n$ = 5; 24 hpi WT $n$ = 7, 24 hpi KO $n$ = 6. *$P$ = 0.0438 (6 hpi), *$P$ = 0.0239 (24 hpi) by two-tailed Mann−Whitney. Source data are provided as a Source Data file. **B** Visualization of IBCs (arrows) in bladder sections collected 6 hpi. Left panels stained with H & E. Right panels stained with a primary antibody to keratin 5 (K5, green) and DAPI. UPEC-RFP appear in red. Scale bars = 100 μm.

(Fig. S7B). Next, we performed a series of in vitro UPEC infection experiments in cultured urothelial cells treated with CsnB. In the first experimental setup, CsnB was applied to cultured 5637 urothelial cells at the same time as UPEC inoculation (Fig. 4A, "Concurrent"). Then UPEC initial binding to cells after 30 min, as well as intracellular infection 2, 4, and 6 hpi, were measured. Aligned with the HA titer results demonstrating no effect of CsnB on type 1 pili, CsnB did not affect UPEC binding to urothelial cells (Fig. S7C). However, gentamicin protection assays to enumerate intracellular bacteria demonstrated that concurrent CsnB treatment caused a dose-dependent decrease in intracellular UPEC CFU 2, 4 and 6 hpi (Figs. 4B and S8A). Next, we determined whether CsnB would still be effective if it were administered to the cells after the 30-minute UPEC invasion phase (Fig. 4C, "Post-invasion"). Under these conditions, the highest dose of CsnB reduced intracellular UPEC (Fig. 4D). These data demonstrated that CsnB treatment is effective at reducing intracellular UPEC infection burden.

**CsnB-limited intracellular UPEC infection is not explained by loss of urothelial cell viability**
Next, we interrogated the mechanism by which CsnB inhibited UPEC infection of urothelial cells. Since Nur77 is known to drive apoptosis and pyroptosis[30,35], we hypothesized that the decreased intracellular bacterial burden could be explained by CsnB triggering urothelial death. Therefore, we measured LDH release to assess urothelial cell viability following treatment with CsnB. CsnB treatment alone for 2 or 4 h did not cause a significant decrease in cell viability; by 6 h of treatment there was a slight, yet statistically significant, decrease in viability only with the highest CsnB dose (vehicle, 100% viable vs. 100 μM CsnB, 89% viable) (Fig. S9A). These data demonstrated that CsnB does not strongly affect 5637 cell viability at the time points employed in our intracellular infection experiments. Since there is potential for synergistic effects of the inhibitor and UPEC infection, we likewise measured cell viability following UPEC inoculation using our "Concurrent" CsnB treatment experimental timeline. Cell viability was >96% for all conditions and time points tested, including as late as 24 h, while treatment with the apoptosis-inducing agent staurosporine as a positive control significantly reduced cell viability, killing all of the cells (Fig. S9B, C). As a final confirmation that induction of programmed cell death was not responsible for the decrease in UPEC intracellular infection, we observed in Fig. 4, we used the pan-caspase inhibitor Z-VAD-FMK. Co-administration of Z-VAD-FMK significantly inhibited staurosporine-induced urothelial cell death (Fig. S9C) but did not affect the ability of CsnB to reduce UPEC CFU (Fig. S9D). Taken together, these data demonstrate that loss of urothelial cell viability is not the mechanism by which CsnB inhibits UPEC intracellular infection in vitro.

**CsnB triggers expulsion of UPEC from infected urothelial cells**
Urothelial cells can rapidly expel internalized UPEC back to the cell exterior via a process involving Rab27b[16,18]. We next tested whether enhanced expulsion was a mechanism by which CsnB treatment reduced intracellular UPEC. We measured UPEC expelled from urothelial cells using a previously published assay[17]. As in our initial experiments, UPEC was added to urothelial cells for 30 min, and then were washed to remove non-adherent bacteria. However, for expulsion experiments, instead of adding gentamicin, the post-wash medium contained a bacteriostatic antibiotic and mannoside to prevent replication and reattachment of expelled UPEC. UPEC expulsion was calculated as the percentage of bacteria present in the medium at each time point relative to the number of bacteria that were intracellular at the end of the 30-minute invasion period ("initial load"). UPEC expulsion was significantly enhanced by CsnB, both in our "concurrent" treatment (Fig. 5A, B) and our "post invasion" treatment models (Fig. 5C, D). Concurrent treatment with the highest CsnB dose resulted in the expulsion of nearly all the initially invaded UPEC (Fig. 5B). The effect of CsnB in the post-treatment model was more modest. The highest concentration induced expulsion of 15% of initially invading UPEC from urothelial cells 4 hpi (Fig. 5D). These data indicate that CsnB can trigger UPEC expulsion.

**CsnB limits intracellular UPEC infection of urothelial cells by blocking endocytosis**
Post-treatment with the highest dose of CsnB decreased intracellular UPEC infection by more than 50% within 2 hpi (Fig. 5B), which was earlier than when expulsion was detectable. Therefore, we suspected that expulsion is not the only mechanism by which CsnB limits UPEC intracellular infection. Consistent with this idea, in the expulsion experiments, we made the surprising observation that the "initial load" of UPEC intracellular CFU 30 min after inoculation, which measures UPEC invasion (Fig. 6A), was significantly lower in CsnB-treated cells compared to controls (Figs. 6B and S8B,C). Treatment with 10 μM

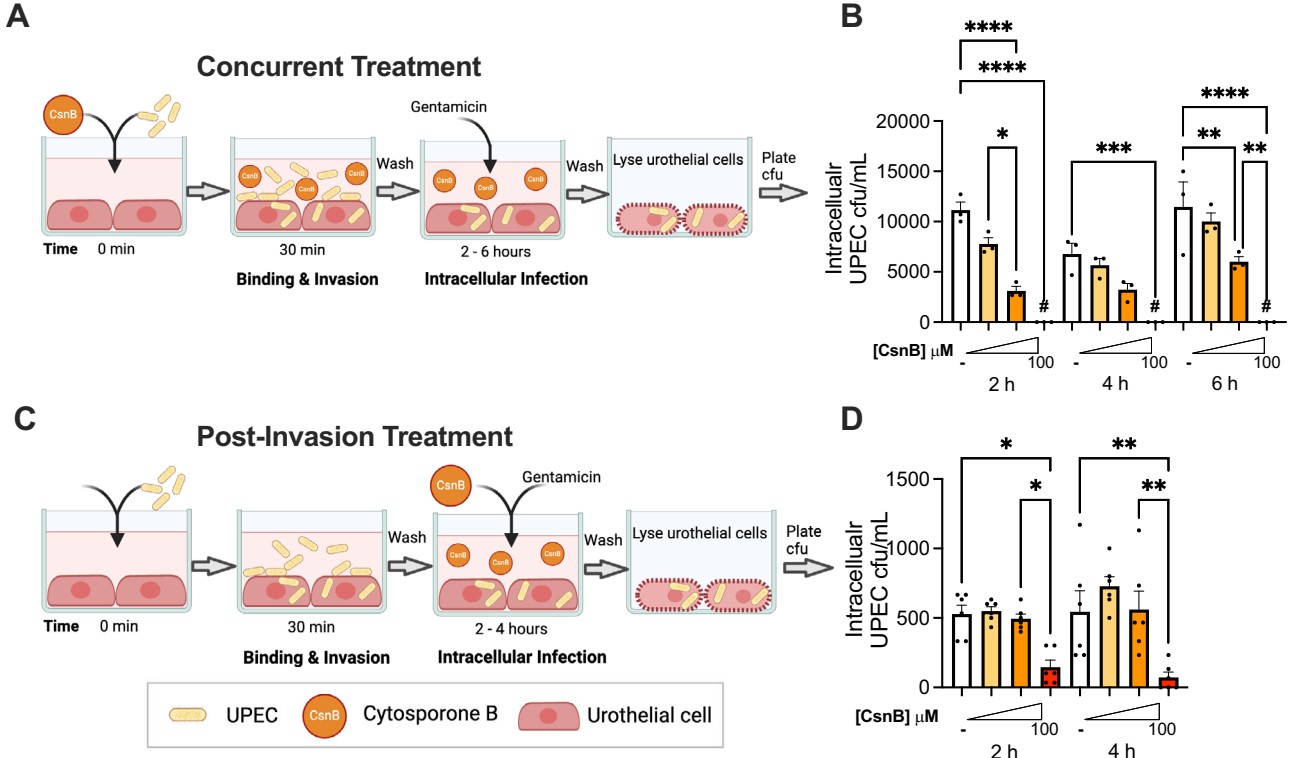

**Fig. 4 | Cytosporone B inhibits UPEC intracellular infection of 5637 urothelial cell line in vitro. A** Schematic of "Concurrent Treatment" experiments in which 5637 cells were treated with DMSO vehicle control (−) or increasing concentrations of CsnB and infected with UPEC at the same time. **B** Time course of intracellular UPEC titers in the Concurrent Treatment model. # denotes no detectable CFU. Data are from $n = 3$ biological replicates per experimental condition (see Fig. S8A for additional independent experiment). ****$P < 0.0001$, ***$P = 0.0004$, **$P = 0.0053$ (6 hpi DMSO vs 10 μM), **$P = 0.0019$ (6 hpi 10 vs 100 μM), *$P = 0.0240$ by one-way ANOVA with Šídák's multiple comparisons test. **C** Schematic of "Post-invasion Treatment" experiments in which 5637 cells were first infected with UPEC for 30 min to allow invasion to occur, and then treated with vehicle or CsnB. **D** Time course of intracellular UPEC titers in the Post-invasion Treatment model. Each experimental condition includes $n = 6$ biological replicates from two independent experiments (each with $n = 3$ biological replicates). **$P = 0.0026$ (4 hpi DMSO vs 100 μM), **$P = 0.0017$ (4 hpi 10 vs 100 μM), *$P = 0.0232$ (2 hpi DMSO vs 100 μM), *$P = 0.0494$ (2 hpi 10 vs 100 μM), by one-way ANOVA with Šídák's multiple comparisons test. Graphs in (**B**, **D**) show the mean with SEM. **A**, **C** Created with BioRender.com released under a Creative Commons Attribution-NonCommercial-NoDerivs 4.0 International license. Source data are provided as a Source Data file.

CsnB reduced UPEC invasion by 50% and 100 μM CsnB reduced invasion by >90% relative to vehicle-treated cells vehicle-treated cells (Fig. 6C).

The ability of CsnB to block initial UPEC invasion of urothelial cells suggested that it could be inhibiting endocytosis. Previous studies have shown that UPEC invasion can be blocked by a variety of endocytosis inhibitors[47]. To determine whether CsnB interferes with endocytosis, we measured the uptake of fluorescently labeled transferrin (Tf) into urothelial cells in the absence and presence of CsnB. Urothelial cells were incubated with Tf for 5, 10 or 15 min and then acid washed to remove extracellular fluorescent signal. Intracellular Tf was detected by flow cytometry. There was a time-dependent increase in Tf uptake in control urothelial cells treated with DMSO (vehicle) (Fig. 6D), reaching 80% positivity by 15 min (Fig. 6E). In contrast, cells treated with CsnB had significantly less Tf uptake at each time point (Fig. 6D), with <20% positivity at 15 min (Fig. 6E). Together, results from in vitro experiments suggest a multi-pronged inhibitory effect of CsnB on UPEC intracellular infection, both enhancing expulsion and limiting invasion by blocking endocytosis.

**A Nur77 agonist inhibits UPEC IBC formation and persistence in the mouse bladder**

Our in vitro results supported the idea that CsnB could be a novel treatment to limit UPEC invasion and intracellular infection in the bladder. Since results from Nur77-KO mice demonstrated that UPEC formed more IBCs in the absence of Nur77, we hypothesized that activation of Nur77 by CsnB would reduce IBC numbers. Therefore, we treated mice intraperitoneally with 5 mg/kg CsnB[46] beginning 1 day prior to UPEC inoculation (Fig. 7A). Since the effects of CsnB on the bladder have not been previously investigated, we performed H&E staining of bladder sections from mice at relevant time points in our treatment model. Importantly, and consistent with our 5637 cell viability data, histological analysis did not detect bladder tissue damage or urothelial disruption in bladders from mice treated with CsnB (Fig. S10). Consistent with our hypothesis, the number of IBCs in the urothelium 6 hpi was significantly lower in mice that received CsnB (Fig. 7B). The inhibition of intracellular infection by CsnB at the early 6 h time point translated into reduced infectious burden later. By 1 wpi, all mice treated with CsnB had cleared UPEC bacteriuria (Fig. 7C) and harbored significantly lower bladder bacterial burdens, with more than a 3-log reduction in UPEC CFU compared with vehicle-treated mice (Fig. 7D). We noticed that the level of UPEC infection in this experiment was higher than we previously observed at this time point (Fig. 1), perhaps caused by the stress induced by additional manipulations needed for multiple i.p. injections. Therefore, we repeated the experiment and reduced the number of injections to ensure the reproducibility of the effect of CsnB. We also included Nur77-KO mice to confirm that the effect of CsnB was specific to Nur77 (Fig. 7E). CsnB treatment administered from 24 h prior

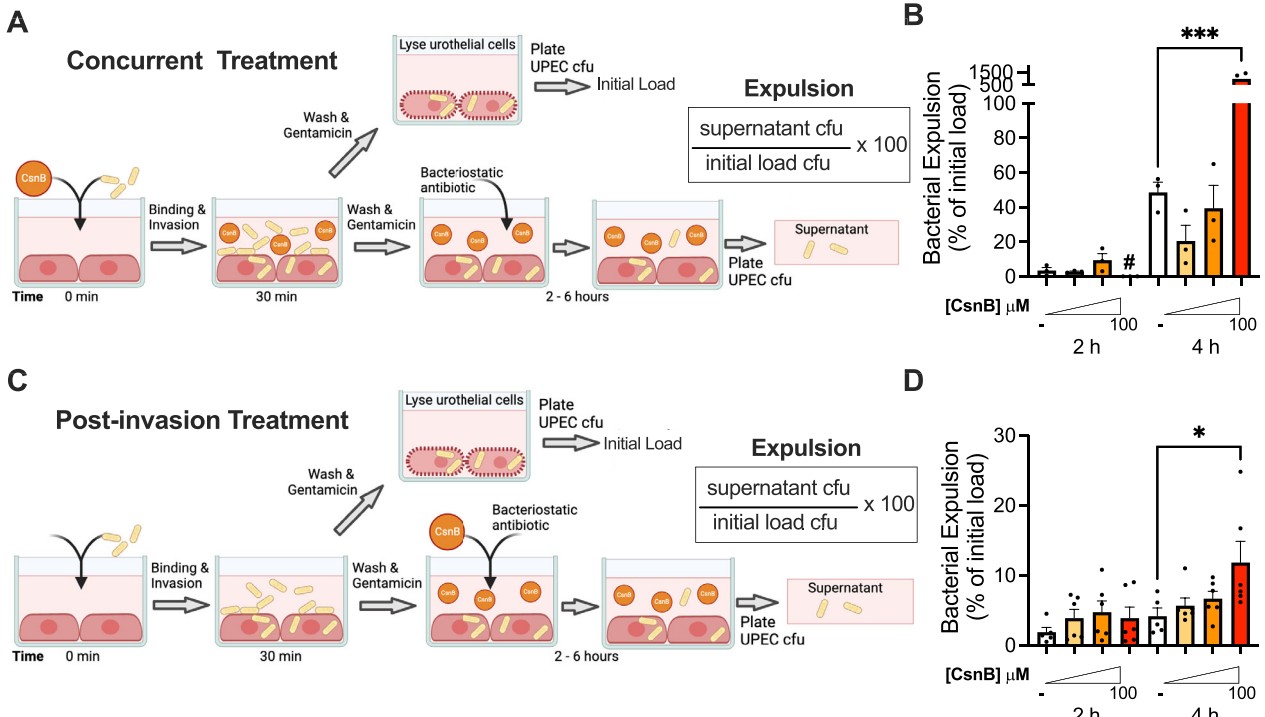

**Fig. 5 | CsnB induces UPEC expulsion from 5637 urothelial cell line. A** Schematic of "Concurrent Treatment" experiments in which 5637 cells were treated with DMSO vehicle control (−) or increasing concentrations of CsnB and infected with UPEC at the same time. UPEC CFU were enumerated in the supernatant and expulsion was calculated relative to the initial load of intracellular UPEC enumerated in parallel wells that were lysed after the 30 min invasion period. **B** Time course of UPEC expulsion from cells treated concurrently with CsnB. # denotes no detectable CFU. Data are from *n* = 3 biological replicates per experimental condition. **C** Schematic of "Post-invasion Treatment" experiments in which 5637 cells were treated with DMSO vehicle control (−) or increasing concentrations of CsnB only after the 30 min invasion period. **D** Time course of UPEC expulsion from cells in the post-invasion treatment model. Each experimental condition includes *n* = 6 biological replicates from two independent experiments (each with *n* = 3 biological replicates). Both graphs denote the mean with SEM. $*P = 0.0146$, $***P = 0.0003$, by one-way ANOVA with Šídák's multiple comparisons test. Source data are provided as a Source Data file. **A, C** Created with BioRender.com released under a Creative Commons Attribution-NonCommercial-NoDerivs 4.0 International license.

to 24 h post UPEC inoculation caused a threefold reduction in UPEC bladder bacterial loads in WT mice 1 wpi. Importantly, this effect was not seen in Nur77-KO mice, which had no significant titer difference between the CsnB and control-treated groups (Fig. 7F). These results complement the data from Nur77-KO mice shown in Fig. 4 and further support the conclusion that Nur77 controls the development of UPEC IBCs and persistent bladder infection.

## Discussion

The results of this study demonstrate that Nur77 is an important mediator of host resistance to UTI. When Nur77 is absent, the bladder is more permissive to intracellular UPEC infection, both acutely and after resolution of bacteriuria. Pharmacological activation of Nur77 in vitro and in vivo yields complimentary results, decreasing UPEC urothelial intracellular infection and enhancing clearance from the bladder. The effectiveness of the Nur77 agonist in vivo identifies Nur77 as a promising therapeutic target for UTI. An increasing number of compounds targeting Nur77 are being developed and beneficial effects are being reported for cancer and inflammatory diseases[39,42,48]. The agonist used here, CsnB, is an octaketide isolated from an endophytic fungus that was the first identified natural ligand for Nur77[41]. CsnB has a strong affinity for Nur77 and not only stimulates the transactivational activity of Nur77, leading to upregulation of Nur77-responsive genes, including Nur77 itself[41], but also triggers nuclear export of Nur77. When Nur77 translocates to the mitochondria, it is a potent death promoter. It is likely that the effects of CsnB on UPEC UTI involve multiple of these Nur77 functions. Our in vitro data demonstrated that CsnB treatment inhibits UPEC invasion of urothelial cells. The

observation that mice treated with CsnB harbored fewer UPEC IBCs in the bladder suggests that CsnB has a similar effect on UPEC urothelial interactions in vivo. Neither type 1 pili nor UPEC binding were affected by CsnB, suggesting that CsnB treatment either prevented uptake of UPEC or rapidly triggered their expulsion. CsnB did not substantially affect urothelial cell viability in our in vitro assays, suggesting that cell death is not required for the agonist to affect UPEC intracellular infection. Even though histological analysis did not reveal apparent disruption of the urothelium, we acknowledge that we cannot exclude the possibility that CsnB-induced apoptosis is involved in vivo. CsnB has been shown to modulate glucose levels, which are also known to affect UTI outcomes. Intraperitoneal injection with 50 mg/kg CsnB (ten-times higher dose than used in our study) increase blood glucose levels 30 min after i.p. injection, but this effect wanes by 2 h. Mice with high blood glucose (streptozocin-induced diabetic model) have increased susceptibility to UTI, with higher bacterial burdens 6, 24, and 72 hpi. Since mice treated with CsnB had decreased susceptibility to UTI, it is unlikely that glucose levels are playing a substantial role in our model. Future studies examining subcellular localization of Nur77 in urothelial cells over the course of UPEC infection and examining the effects of CsnB on the bladder mucosa will distinguish which functions of Nur77 are important for mediating urothelial resistance to UPEC. The data presented here have identified a new host factor involved in urothelial uptake and expulsion of UPEC. To our knowledge, involvement of Nur77 in vesicle trafficking has not been reported; whether the effect of Nur77 on this process is direct or indirect remains to be determined. The finding that UPEC invasion and expulsion can be manipulated by the timing of CsnB treatment provides a new tool that

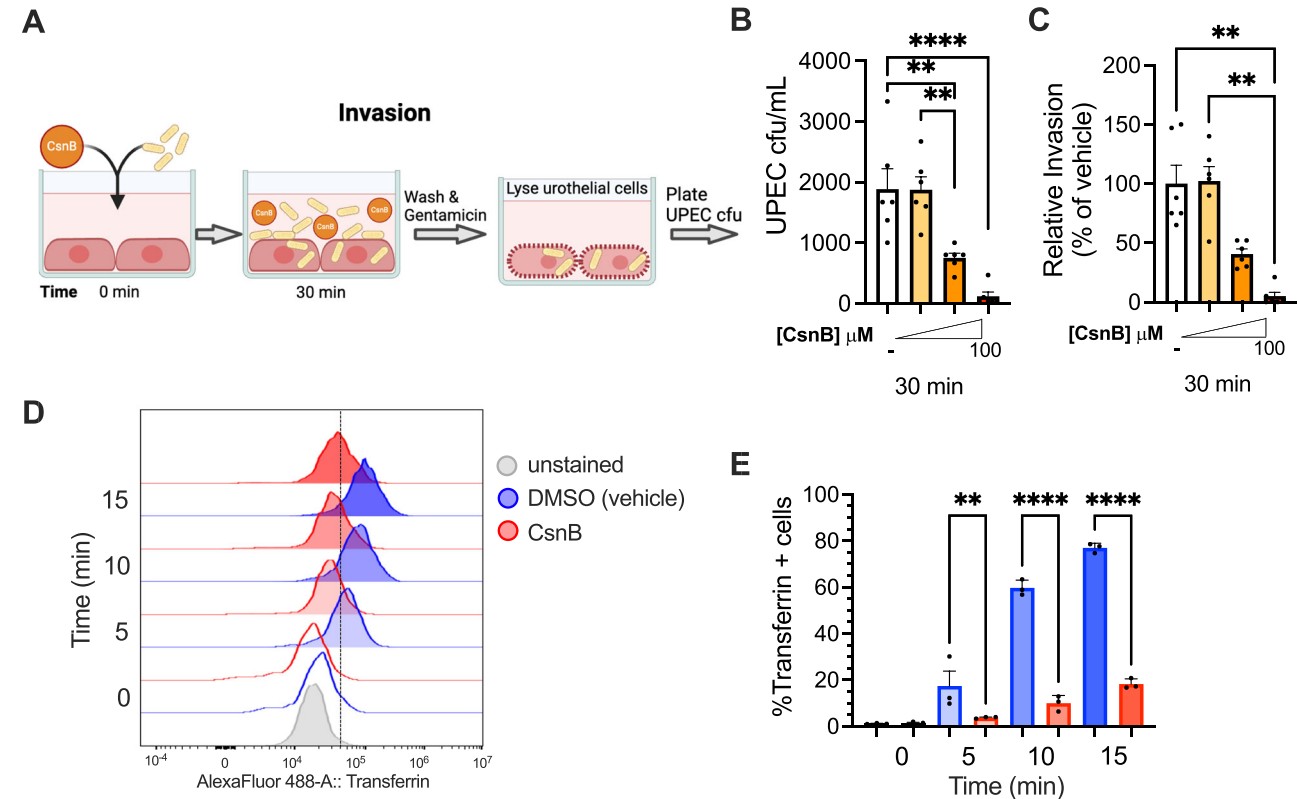

**Fig. 6 | Cytosporone B inhibits UPEC invasion and blocks endocytosis in 5637 urothelial cell line in vitro. A** Schematic of 5637 cell invasion assay. **A** Created with BioRender.com released under a Creative Commons Attribution-NonCommercial-NoDerivs 4.0 International license. **B** Intracellular UPEC titers 30 min post infection in cells treated concurrently with DMSO vehicle control (−) or increasing concentrations of CsnB. ****$P < 0.0001$, **$P = 0.0046$ (DMSO vs 10 μM), **$P = 0.0050$ (1 vs. 10 μM) by one-way ANOVA with Šídák's multiple comparisons test. **C** Relative UPEC invasion 30 min post inoculation, calculated from the CFU data in (**B**). Each experimental condition in (**B**, **C**) includes six biological replicates from two

independent experiments. **$P = 0.0019$ (DMSO vs 100 μM), **$P = 0.0014$ (1 vs 100 μM) by Kruskal–Wallis with Dunn's multiple comparisons test. **D** Histograms of a time course of intracellular transferrin detected by flow cytometry. The dotted line denotes gating for defining transferrin + cells in (**E**). **E** Time course of transferrin positivity expressed as % of live cells in each condition. ****$P < 0.0001$, **$P = 0.0065$ by one-way ANOVA with Šídák's multiple comparisons test. **D**, **E** Are representative to two independent experiments, each with three biological replicates for each condition and time point. Bars denote the mean with SEM. Source data are provided as a Source Data file.

will be useful in studies of host mechanisms modulating the UPEC intracellular infection cascade. It is possible that Nur77 activation could likewise be important for the intracellular lifestyle of other pathogens.

The data from mouse UTI experiments pointed to Nur77 exerting its effect in the urothelium, rather than via myeloid or lymphoid cells. This conclusion is further corroborated by infection experiments in urothelial cells in vitro. Our observations that WT and Nur77-KO displayed similar immune cell populations in the bladder 24 h after UPEC infection are consistent with the results from an *E. coli* peritonitis model that displayed no differences in inflammatory cytokine levels, neutrophil or macrophage numbers in WT and Nur77-KO mice in all body compartments tested[49]. In addition, isolated peritoneal macrophages from WT and Nur77-KO mice showed no differences in cytokine expression patterns in response to *E. coli*[49]. Surprisingly few studies have examined the role of Nur77 in bacterial infections. Nur77 is known to be expressed by gut and lung epithelial cells but has almost exclusively been studied in the context of noninfectious diseases such as cancer. Perhaps the most relevant evidence suggesting that Nur77 could participate in host responses to intracellular bacterial infection comes from the recent discovery that Nur77 is a cytosolic sensor for intracellular LPS[35]. The endogenous ligand(s) for Nur77 were unknown until very recently. Biochemical screening discovered that Nur77 is an LPS-binding protein[35]. Additional data demonstrated that Nur77 binding of LPS and of mitochondrial DNA leads to activation of the non-canonical NLRP3 inflammasome[35]. UPEC are known to activate the

NLRP3 inflammasome in urothelial cells in an NFκB-dependent manner[50]. We reason that the LPS-sensing function of Nur77 could be particularly important for host recognition of UPEC urothelial invasion during UTI. Our results suggest that Nur77 should be further examined as a host sensor of bacterial infections, especially those at mucosal surfaces and by pathogens that invade epithelial cells.

This study has some limitations that provide opportunities for future research. First, we used available Nur77-KO mice that lack Nur77 in all cell types. Future generation of urothelial-specific Nur77-KO mice would allow more definitive assessment of the contribution of Nur77 expression in urothelial cells to the host response to UTI in vivo. A second limitation is that our flow cytometry analysis of the mouse bladder examined immune cell populations generally and did not assess activation states or sub-populations. Therefore, it remains possible that certain immune cells contribute to the overall phenotypes observed in Nur77-KO mice. For example, we did not distinguish between resident and circulating monocyte-derived macrophages, which are known to have distinct functions in the bladder response to UTI[51]. We also did not examine innate lymphoid cells (ILCs), which are largely unstudied in the bladder but may serve yet unknown functions in UTI responses. Future studies examining immune populations in Nur77-KO bladders that are known to be important for UTI, such as M1-like and M2-like macrophages[52] and tissue-resident T cells[53] will further illuminate the contribution of Nur77 to immune responses to, and control of, UTI. Although here we examined Nur77 only during the initial UTI, it is likely that Nur77 influences adaptive immune responses

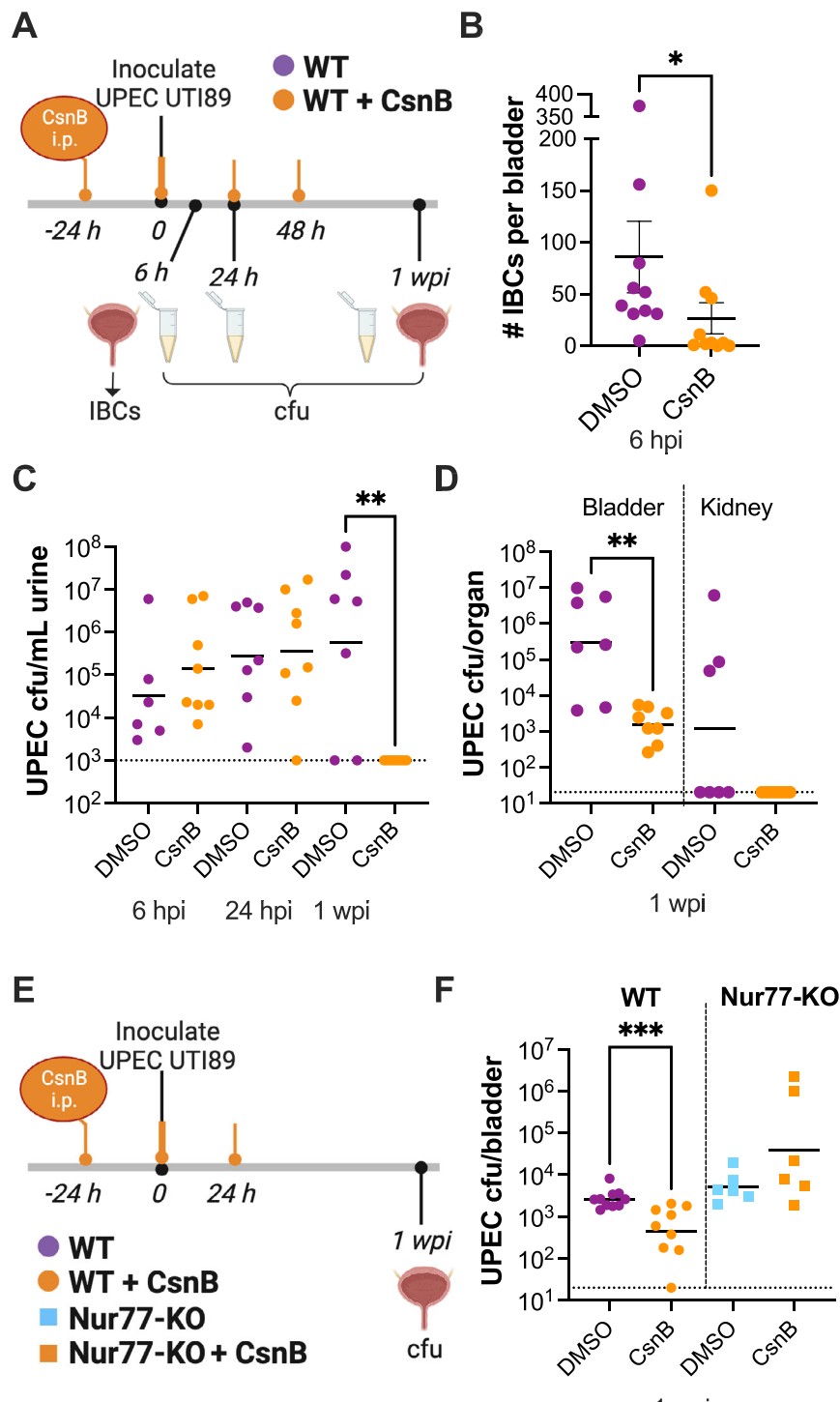

**Fig. 7 | Mice treated with CsnB are protected from UPEC bladder infection.**
**A** Schematic of CsnB treatment and UPEC UTI mouse model for data in (**B**–**D**).
**B** Enumeration of IBCs in splayed bladders 6 hpi. $n = 10$ mice per group. Results are combined from two independent experiments. *$P = 0.0198$ by Mann–Whitney test.
**C** Time course of UPEC titers in urine. **$P = 0.0016$ by Kruskal–Wallis with Dunn's multiple comparisons test. **D** UPEC titers in bladder and kidney tissue homogenates 1 wpi. **$P = 0.0036$ by Mann–Whitney test. **C**, **D** DMSO-treated $n = 7$, CsnB-treated

$n = 8$. **E** Schematic of CsnB treatment and UPEC UTI mouse model in WT (DMSO-treated $n = 10$, CsnB-treated $n = 9$) and Nur77-KO ($n = 6$ per group) mice for data in (**F**). **F** UPEC titers in bladder tissue homogenates. Results are combined from two independent experiments. ***$P = 0.0007$ by Mann–Whitney test. Graphs in (**C**, **D**, **F**) denote geometric mean values. Graph in B shows mean +/− SEM. Source data are provided as a Source Data file. **A**, **E** Created with BioRender.com released under a Creative Commons Attribution-NonCommercial-NoDerivs 4.0 International license.

in the bladder. Nur77 is a well-established mediator of T-cell function, being first discovered for its role in triggering apoptosis during T-cell-negative selection[54]. It is an early activation marker of TCR signaling[55], and a Nur77-GFP reporter mouse is frequently used to monitor T-cell

activation in vivo[56]. Nur77 is used as a reporter gene for antigen receptor signaling in T cells and iNKT cells[56,57]. Nur77 is an important mediator of peripheral B cell tolerance[58]. Studies in mice have demonstrated that adaptive immunity mediated by T and B cells is

established in the bladder following clearance of an initial UPEC UTI[40]. Future studies using the well-established UPEC challenge infection model in Nur77-KO mice are needed to uncover additional roles for Nur77 in adaptive immunity to UTI. UTI are highly recurrent, with up to 50% of women experiencing rUTI in their lifetime[1]. A better understanding of what host factors control adaptive immune responses to UTI could reveal new therapeutic avenues to prevent recurrences. Finally, our data demonstrate efficacy of pharmacological targeting of Nur77 given prophylactically to decrease UTI severity by limiting UPEC intracellular invasion and enhancing its clearance from the bladder. Further studies are needed to discover whether a Nur77 agonist would be an effective treatment if given after UTI onset or to eliminate established bladder reservoirs.

In summary, this study establishes that Nur77 is an important host factor controlling intracellular UPEC infection in the bladder. We propose that pharmacological targeting of Nur77 should be explored as a new therapeutic approach for UTI. Since such an approach would target a host protein, it would not engender resistance among uropathogens and should be agnostic to antimicrobial resistance in multiple organisms, especially those which are known to invade urothelial cells, including *E. coli*, *Klebsiella pneumoniae*[59,60], *Pseudomonas aeruginosa*[61], *Acinetobacter baumannii*[25], and *Enterococcus faecalis*[62]. These uropathogens are responsible for substantial clinical morbidity and mortality in the community and in hospitalized patients, an increasing number of whom have few antibiotic options. Thus, the public health significance of a new efficacious approach targeting a host factor to help eliminate uropathogens from the bladder, as proposed here, is vast.

## Methods
All animal procedures were performed with the prior approval of the Washington University Institutional Animal Care and Use Committee (protocols #20-0031 and 23-0015).

### Study design
The objective of our study was to assess the contribution of the mammalian orphan nuclear receptor Nur77 (aka NR4A1) to urinary tract infection. For this, we assessed outcomes of experimental UTI with UPEC, profiled immune cell populations, and analyzed urothelial architecture in C57BL/6 mice lacking Nur77 (Nur77-KO) compared to wild-type control mice. We examined Nur77 expression in published single-cell RNAseq datasets from bladder cells isolated from both mice and humans. We also used the small-molecule Nur77 agonist cytosporone B (CsnB) with the urothelial 5637 cell line and in vivo infection models to understand the mechanism by which Nur77 interferes with UPEC intracellular infection. Each in vivo experiment was performed at least in duplicate. Intermediary and final endpoints were predetermined for all experiments. Mouse numbers for experiments assessing UTI were predetermined based on our prior published data on UPEC titers in urine and tissue[24,63,64]. For flow cytometry, a pilot study of mice was conducted to reduce the total number of animals used, estimate variability among animals, and evaluate procedures. Mouse inoculations generally were not blinded, but investigators performing immunofluorescence microscopy and flow cytometry were blinded to the experimental groups during data acquisition. Age-matched WT and Nur77-KO mice were randomly allocated to mock or UTI89 infection and/or to vehicle or CsnB treatment groups. Each in vitro experiment was performed at least in duplicate, with at least six biological replicates per condition and time point. The number of samples and the number of experimental replicates for each experiment are reported in figure captions. All data points are included in each figure.

### Mice
All mice were group-housed in a temperature-controlled room in a specific pathogen-free facility with a 12 h light/12 h dark cycle. Mice were given water and standard chow diet ad libitum. Nur77-deficient mice were purchased from Jackson Laboratories (B6;129S2-Nr4a1tm1Jmi/J #006187) and bred in house thereafter. Age-matched C57BL/6J mice (Jackson Laboratories; #000664) were purchased and housed in our facility a minimum of seven days prior to experimentation. We compared Nur77-KO mice to wild-type C57BL/6J to be consistent with the majority of published studies that have used the Nur77-KO strain. An alternative option for future studies would be to use the B6129PF2/J that is an F2 hybrid from C57BL/6J females (B6) and 129P3/J males (129P) approximate controls for genetically engineered strains that were generated with 129-derived embryonic stem cells, which is how the Nur77-KO strain was generated. In order to utilize the well-established model of transurethral inoculation, and because UTI predominantly affect women, all experiments only used female mice.

### Bacterial strains and preparation
Uropathogenic *E. coli* strains UTI89, harboring a kanamycin resistance cassette[65], or UTI89-RFP[40] (kind gift from Molly Ingersoll) were grown aerobically at 37 °C in static liquid culture in Lysogeny Broth (LB) medium, or on LB agar plates with 50 μg/ml kanamycin. Mouse inocula containing $1 \times 10^7$ CFU UPEC in 50 μL PBS (OD = 0.35) were prepared for each experiment from static LB liquid cultures[63].

### Mouse urinary tract infection and cytosporone B treatment
Experiments were performed essentially as described previously[24,63]. Seven-week-old female wild-type C57BL/6 J and Nur77-KO mice were anesthetized and then transurethrally inoculated with $1 \times 10^7$ CFU UPEC in 50 μL PBS. Urine was collected throughout the course of infection (6 and 24 hpi and weekly thereafter) to measure bacteriuria. In some experiments, either 0.1 mg CsnB (Tocris, Cat. No. 5459) or vehicle (DMSO) in 100 μL PBS (5 mg/kg final dose[46];) was given IP (intraperitoneal) at the indicated time points pre- and post infection. Injections were freshly prepared by diluting a CsnB stock solution (reconstituted in DMSO) or an equal volume of DMSO as vehicle control. At the time of sacrifice, bladders and kidneys were harvested, homogenized in 1 mL sterile PBS and plated on LB plates containing 50 μg/mL kanamycin for CFU measurements.

### Nur77 expression analysis
We obtained Nur77 expression data from two previously published datasets[37,38]. t-distributed stochastic neighbor embedding (t-SNE) plots of annotated bladder cell types, t-SNE plots of Nur77-expressing bladder cells, and violin plots of Nur77 expression levels in C57BL/6 mouse bladders were produced from https://tabula-muris.ds.czbiohub.org. Average Nur77 expression data in bladder cells were obtained from ref. 38 and are available in our Source Data file. Data were plotted in GraphPad Prism.

### Flow cytometry
Bladders were aseptically harvested postmortem and digested enzymatically as described in ref. 63 for the 24 hpi (Fig. 3B–F) experiments. In short, the bladders were emptied, placed in 100 μL Tyrode's solution (140 mM NaCl, 5 mM KCl, 1 mM MgCl₂, 10 mM D-Glucose and 10 mM HEPES, pH 7.4), and minced with scissors. Digestion buffer containing 5 mg BSA, 0.03 mM CaCl₂, 132.5 units Collagenase Type 1, 96.4 units Collagenase Type III, 50 units Collagenase Type VI, 10 units DNase, 115 units Papain, 50 units Pan Collagenase and 10.5 units Hyaluronidase in Tyrode's Solution was added to the bladders and incubated at 37 °C on a nutator for 40 min. The samples were then pipetted up and down for one minute to further separate the bladder cells and centrifuged for 10 min at 350×g at 4 °C. The supernatant was discarded, the pellet was resuspended in 1 mL Accutase (Millipore Sigma; #A6964) to detach any cells adhering to the tube and nutated for 10 min at 37 °C. For naive (Fig. S2) and 4 wpi (Fig. 3H–M) experiments, aseptically harvested bladders were digested with liberase and DNase as described in ref. 43.

Following digestion, the samples were centrifuged and the pellets were resuspended in RBC cell lysis buffer (155 mM $NH_4Cl$, 10 mM $KHCO_3$) for 1 min at room temperature and 9 mL of PBS was added to stop lysis. The samples were then filtered through a 70-μm filter and any remaining tissue was manually disrupted to make a final cell suspension. The samples were then centrifuged and the pellets were resuspended in 1 mL PBS. Cell counts were obtained via hemacytometer and $1 \times 10^6$ cells were added to FACS tubes. The dead cells were stained with Live/Dead Fixable Blue (Thermo Fisher Scientific; #L34962) and incubated on ice for 30 min in the dark. The cell suspension was washed with FACS buffer (PBS containing 0.01% FBS and 0.1% sodium azide), centrifuged, and incubated with Fc Block (BD Pharmingen; #553142) for 15 min at 4 °C. Following the incubation, antibody cocktail containing CD45 BV510 (BD Horizon; #563891), MHCII BUV395 (BD Optibuild; #743876), CD19 BUV661 (BD Horizon; #612971), CD3 BUV805 (BD OptiBuild; #741982), CD11b BV570 (Biolegend; #101233), F4/80 BB700 (BD Optibuild; #746070), NK1/1 PE/Dazzle594 (Biolegend; #108748), Ly6C PE-Cy5.5 (Novus Biologicals; #N100-65413PECY55), Ly6G AF700 (Biolegend; #127621), Siglec F AF421 (Biolegend; #155509) was added and incubated for 45 min at 4 °C covered from light. After antibody staining, the cells were washed two times in FACs buffer, filtered through 40-μM nylon mesh and analyzed on the Cytek® Aurora (Cytek Biosciences) flow cytometer. Blood was collected via cardiac puncture at the time of sacrifice and joined the bladders at the red blood cell lysis step. All flow analysis was performed using FlowJo (Treestar) software. Immune cell populations were identified according to our established gating strategy (Fig. S11).

## Microscopy

Bladders were harvested at the specified time points and fixed in 10% formalin at room temperature overnight and then transferred to 70% ethanol. Bladders were embedded in paraffin and sagittal sections were prepared and mounted on glass slides. For histological analysis, slides were stained with hematoxylin and eosin (H&E) according to standard protocol. For immunofluorescence microscopy, slides were placed onto glass holding trays and then placed into fresh Histo-Clear® Histological Clearing Agent, (National Diagnostics), for 10 min two times. The trays were drained, then moved to 100% ethyl alcohol for 10 min two times, and then to 95% ethyl alcohol two times for 10 min. Finally, glass trays holding the slides were placed under running water for 10 min. During the deparaffinization, fresh pH 9 and 6 buffered antigen retrieval solutions were made and brought to a boil in 50 ml BD conical tubes in a glass beaker filled with water in a steamer. After washing, glass slides were placed into the appropriate buffer around 90–100 °C without allowing the slides to dry out and boiled for 15 and 30 min in for pH 9 and pH 6 buffer, respectively. Slides were then cooled and allowed to cool to 60 °C and were washed in 0.5% Triton X-100 in phosphate-buffered saline (PBS) at room temperature. Glass slides were removed from wash buffer one at a time, placed horizontally into humidified slide boxes, and the hydrophobic boundary was marked around the perimeter with a PAP PEN. About 300 μl of 10% heat-inactivated horse serum (HIHS) and 3% bovine serum albumin (BSA) in 0.5% Triton X-100 PBS were placed onto each slide for blocking. Slides were incubated in a closed, humidified slide box for 1 to 2 h. During blocking, primary antibody cocktails of chicken anti-KRT 5 1:500; goat anti-P63 1:300; mouse anti-Upk2 1:50 were prepared in 1% HIHS and 1% BSA in 0.5% Triton X-100 sufficient for around 300 μl per slide. The blocking solution was removed by vacuum and the primary antibody added onto the slide without disturbing hydrophobic perimeter. Slides were incubated overnight in humidified slide boxes at 4 °C. The following day, the primary antibody was removed, and the slides were washed for 10 min in fresh 0.5% Triton X-100 PBS twice. During washes, secondary antibody cocktails were prepared in 1% HIHS and 1% BSA in 0.5% Triton X-100 for 300 μl per slide, and two drops of NucBlue® nuclear-staining reagent (DAPI) were added per milliliter of antibody cocktail. Slides were removed from the washing buffer and hydrophobic perimeters redrawn then a secondary antibody cocktail was added. Slides were incubated in the dark at room temperature for 30 min to 1 h. After incubation, slides were washed, and previously warmed DAKO glycerol mounting medium was applied before the coverslip. Slides were stored overnight at 4 °C in slide folders.

## IBC enumeration

Bladders were harvested 6 h and 24 h post infection and splayed out with pins on Slygard® 184 (The Dow Chemical Company; #2646340) cured six-well plates containing PBS. The bladders were then washed with PBS and fixed in 4% paraformaldehyde for 1 h at room temperature. Following fixation, the paraformaldehyde was removed, and the bladders were washed one time with PBS and three times (5 min each) in LacZ wash solution (2 mM $MgCl_2$, 0.01% w/v sodium deoxycholate and 0.02% w/v NP40). After the washes, LacZ stain (LacZ wash solution plus 1 mg/ml X-gal and 1 mg/mL K-ferrOcyanide/K-ferrIcyanide) was applied to each bladder and incubated at 30 °C overnight protected from light. IBCs were quantified by counting the number of X-gal stained (blue) bacterial clusters, which appear as distinct round puncta under a dissecting microscope.

## Bladder epithelial cell infections and CsnB treatments

The human bladder epithelial cells, 5637 (ATCC; #HTB-9), were grown in RPMI-1640 medium supplemented with 10% FBS. At confluency, the cells were plated in a 24-well plate at a density of $2 \times 10^5$ cells/well and incubated overnight at 37 °C, 5% $CO_2$. The following day, the cells were infected with UPEC at a MOI of 100, centrifuged (spinoculated) and incubated for 30 min at 37 °C, 5% $CO_2$. Cells were washed three times with PBS to remove non-adherent bacteria. Bacterial binding was assessed by lysing designated wells with 0.1% Triton X-100 in PBS for 10 min at RT on orbital rocker, serially diluting, and plating on LB+kan plates. The remaining wells were treated with 100 μg/mL gentamicin. Bacterial "initial load" was assessed by lysing infected cells in designated wells after 10 min of gentamicin treatment and three additional PBS washes. Relative invasion was calculated by dividing the intracellular CFU in each well by the average input CFU in the DMSO control wells. At this time, to the expulsion wells were added 100 mM methyl α-mannopyranoside and 25 ug/ml trimethoprim for the remaining specified times. At 2, 4, or 6 h post UPEC exposure, one-tenth of each supernatant was removed and plated on LB+kan plates. The intracellular infection wells were incubated in the presence of gentamicin for the indicated lengths of time and then lysed for CFU enumeration. Cytosporone B (Sigma; #C2997) was given at indicated doses (vehicle only, 1 μM, 10 μM, or 100 μM; concentration range based on previous reports[66,67]) either at the time of UPEC infection (concurrent treatment) or after the gentamicin treatment and washing to remove extracellular bacteria (post-invasion treatment) and continuously throughout the assay. Each condition employed three wells in each assay.

## Targeted knockdown of Nur77 with siRNA transfection

In all, 5637 cells were seeded in 24-well plates at density of $1 \times 10^5$ in RPMI media (supplemented with 10% FBS and 1% penicillin–streptomycin) and maintained at 37 °C in a humidified atmosphere containing 5% $CO_2$. On the day of transfection, twenty picomoles of Nur77-targeting siRNA (SR302153B, ORIGENE, Rockville, MD) and non-targeting siRNA (SR30004, ORIGENE, Rockville, MD) were each combined with diluted Lipofectamine RNAiMAX transfection reagent (Invitrogen Inc., Carlsbad, CA) in 1×Opti-MEM reduced serum medium (without antibiotics). Each mixture was incubated at room temperature for 10 min to allow the formation of siRNA-lipid complexes. The siRNA-lipid complexes were added to each well containing 5637 cells at 60–70% confluency, gently rocked to ensure even distribution and incubate for 24 h at 37 °C in a humidified atmosphere

containing 5% $CO_2$. After incubation, the transfection media was replaced with 1 mL of fresh complete RPMI media, and cells were further incubated for 24 h and then UPEC infection experiments were performed as described above. The efficiency of siRNA-mediated knockdown was assessed by real-time PCR (qRT-PCR) to quantify target mRNA. Primers Nur77 F 5′-GGA CAA CGC TTC ATG CCA GCA T-3,' Nur77 R 5′-CCT TGT TAG CCA GGC AGA TGT AC-3', GAPDH F 5′-CAT CAC TGA CAC CCA GAA GAC TG-3, GAPDH'R 5′-ATG CCA GTG AGC TTC CCG TTC AG-3'. Cycling parameters: 95 °C for 3 min initial denaturation, Denaturation at 95 °C for 10 s Extension at 60 °C for 30 s X40 cycles. qPCR machine Applied Biosystem 7500 Fast Real-Time PCR System.

## Cytotoxicity assays

At the specified time points, cytotoxicity of CsnB-treated 5637 bladder cells was determined by measuring LDH release with the CytoTox 96® Non-Radioactive Cytotoxicity Assay (Promega; #PRG1780) per manufacturer instructions. Absorbance values were recorded on a Tecan Infinite M-Plex plate reader (Männedorf, Switzerland). In each experiment, parallel vehicle-treated cells were lysed with 0.1% Triton to measure maximal LDH release. The % killed cells was calculated relative to the maximum LDH release (absorbance/triton max absorbance) *100, which was then used to calculate % viability (100 − % killed).

## Cytotoxicity of cytosporone B to UPEC

Cytotoxicity of CsnB against UPEC was evaluated under conditions that mimicked those of the urothelial infection experiments. UPEC were grown in LB and an inoculum was prepared in PBS. UTI89 and the indicated doses of CsnB, or DMSO vehicle control, were added to RPMI + FBS media in a 24-well plate and incubated 30 min at 37 °C with 5% $CO_2$. Following incubation, samples were serially diluted and plated to enumerate UPEC cfu.

## Hemmaglutination assay

Type 1 piliation of UPEC was assessed with the standard hemagglutination assay. UTI89 from a $2 \times 24$ h LB static liquid culture was diluted to $OD_{600} = 1$ and 1 mL was spun and the pellet resuspended in 100 μL PBS. The bacterial suspension was mixed initially 1:1 (25 μL each), and subsequently seven additional serial 2-fold dilutions were prepared with PBS with or without 100 μM CsnB in a V-bottom 96-well plate. Fresh guinea pig red blood cells prepared in PBS were added to each well on the plate and incubated for 2 h at 4 °C. HA titers were determined as the maximal bacterial dilution that produced a "haze" in the well.

## Statistical analysis

For mouse and in vitro studies, statistical and graphical analyses were performed using GraphPad Prism software (version 10.0 or earlier). Normality was determined by the Shapiro–Wilk normality test. For comparisons between two groups, unpaired t tests were used on normally distributed samples and Mann–Whitney tests on samples that did not pass the normality test. One-way analysis of variance (ANOVA) was used on normally distributed samples, and Kruskal–Wallis tests were performed on samples that did not pass the normality test. Šídák's multiple comparisons test was performed for CsnB dose-response experiments. All measurements were taken from distinct samples. In Figs. 1B, C and 7C, the measurements correspond to distinct samples taken sequentially over different time points from the same animals as indicated. Results were considered statistically significant at $P < 0.05$ and exact $P$ values are indicated in the figure legends. For bacterial CFU data, bars represent the geometric mean. For flow cytometry data and IBC enumeration, error bars represent mean and SEM unless otherwise noted. For all figures with multiple comparisons across groups, except those involving repeated measurements over time, all significant differences across groups are noted, and a lack of notation indicates a lack of statistical significance.

For figures with multiple comparisons over time, only significant differences relative to controls at the same time point are shown, and lack of notation indicates a lack of significance relative to those controls. Other relevant comparisons across groups were noted as indicated in each figure legend.

## Reporting summary

Further information on research design is available in the Nature Portfolio Reporting Summary linked to this article.

## Data availability

Single-cell RNAseq data shown in Fig. S2 are available in the previously published studies https://tabula-muris.ds.czbiohub.org and ref. 38. All data generated in this study are provided in the Source Data file. Source data are provided with this paper.

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

## Acknowledgements

This work was supported by funding from the NIH National Institute of Diabetes, Digestive, and Kidney Diseases (NIDDK) R03DK132442 (N.M.G.), K01 DK110225 (N.M.G.), U54 DK104309 (C.M.M.), and R01DK137964 (N.M.G.). The authors thank Molly Ingersoll and Brian Becknell for providing us with the UTI89-RFP strain, Hunter Kuhn and Rebekah Smither for technical assistance with LDH and HA assays, Teri Hreha for helpful advice on flow cytometry analysis, and David Hunstad for insightful discussions and thoughtful comments and edits to our manuscript.

## Author contributions

Conceptualization: N.M.G. Methodology: all authors. Investigation: all authors. Visualization: N.M.G., C.A.C., C.W., E.B., and L.K. Funding acquisition: N.M.G. and C.M.M. Project administration: N.M.G. Supervision: N.M.G. and C.M.M. Writing—original draft: N.M.G. Writing—review and editing: all authors.

## Competing interests

The authors declare no competing interests.
