## [Peer Review File · Nature Communications]

REVIEWER COMMENTS

Reviewer #1 (Remarks to the Author):

This study by Gilbert and colleagues examines the effects of the nuclear receptor Nur77 on UTI using knockout mice and a Nur77 agonist coupled with bladder cell culture and mouse infection models. This study is among the first to consider the effects of Nur77 on bacterial infection, which is especially interesting given a recent 2023 report indicating that Nur77 can act as a sensor for cytosolic LPS. A previous 2021 study by the corresponding author laid the groundwork for this current submission, presenting evidence that the deletion of Nur77 protects mice from *Gardenerella*-induced resurgence of UPEC reservoirs. Data presented in this submission indicates that Nur77 can affect UPEC titers within the bladder tissue at later time points beyond 2 wpi, and that more IBCs develop with Nur77 KO mice at early time points. In contrast, the Nur77 agonist CsnB is reported to inhibit UPEC invasion of bladder cells as well IBC numbers, levels of bacteriuria, and bacterial titers associated with the bladder tissue. The authors also show that Nur77 deletion does not have any gross effects on immune cell populations within the bladder during UTI, and provide data suggesting that the absence of Nur77 impairs full regeneration of the urothelium at 4 wpi. Overall, the data indicating that Nur77 impacts the longer-term survival of UPEC within the bladder are convincing. However, there are some discrepancies in the data, as noted below, along with some additional controls/suggestions that might help better resolve the effects of Nur77 on UTI and help strengthen the authors' conclusions. The use of CsnB to reduce UPEC titers in the cell culture and mouse models is an especially interesting aspect of this study, but how CsnB is acting to limit UPEC colonization in these assays could be better delineated.

Specific concerns/suggestions:

Major issues

1. Considering results in Fig 5A showing that mice lacking Nur77 have more IBCs at 6 and 24 hpi, it seems surprising that the KO mice do not have higher bladder titers at earlier time points before 3 wpi (Figs 1D and E). Quantifying all intracellular bacteria (e.g. via ex vivo gentamicin protection assays) at these time points could be very helpful in understanding the effects of Nur77. This would also complement the data in Fig 6G indicating that the agonist CsnB inhibits UPEC invasion of bladder cells.
2. Lines 246-248, the authors suggest that the "decrease in measured intracellular UPEC caused by CsnB was not solely due to Nur77-driven urothelial cell death." To better prove this point, the authors could treat the cells with both CsnB and an apoptosis inhibitor (or some similar approach to help rule out host cell death as a confounding factor). The argument that CsnB-induced host cell death is not responsible for the decreased levels of invasion (Fig 6G) does not work so well when considering CsnB effects on UPEC expulsion (Fig 6F and H), which is maximal when host cell death levels are also high. The conclusion that CsnB/Nur77 activation enhances expulsion should be better justified.

Does CsnB have a general inhibitory effect in endocytotic pathways. For example, will CsnB inhibit uptake of fluorescently-labeled transferrin by 5637 cells?

3. Considering the ability of CsnB to induce death of 5637 cells, the authors should determine if the agonist is also inducing death of urothelial cells in the mouse experiments presented in Fig 7.

Does CsnB inhibit UPEC invasion in vivo, as it does in the cell culture assays?

Other issues

4. The section starting on page 6 that uses publicly available transcriptomics data to assess Nur77 expression in human and mouse bladder tissues could be presented more concisely, and Fig 2 could be supplemental. For this section, what is the difference between “urothelial cell” and “bladder cell”? Independent confirmation by the authors that Nur77 is expressed in the urothelial cells, and especially the superficial cells, would strengthen this study.

5. Reference to “surface” CK20 as a marker of terminally differentiated umbrella cells (line 197, Fig4B), is a bit confusing, as I don’t think this protein is surface-exposed like the uroplakins. “luminal” might be a better descriptive term. Reduced CK20 staining is the primary/only evidence given to support the idea that regeneration of the urothelium is impaired in Nur77 KO mice. Zoomed in images/insets of some of the umbrella cells depicted in Fig 4A would help the reader better appreciate this phenotype. Use of additional differentiation markers would help back up the conclusions from this section. Is expression of the uroplakins, or their assembly into plaques, diminished in the Nur77 KO mice at 4 wpi? Are size or multi-nucleate status of the umbrella cells altered in the KO? Is regeneration also impaired following injury of the urothelium in the absence of infection (e.g. following cyclophosphamide injury)?

6. Lines 264 and 334, 5637 cells do not produce uroplakin plaques and do not have fusiform vesicles. The term “endosomes” is more appropriate in this system.

7. Why are dotted lines around the different groups of figures included in Fig. 6?

8. Caption for Fig.1D indicates that the data are from 1 experiment. This is not standard in the field, is in contrast to what is stated in the Methods, and different from the other mouse experiments where results from at least two independent experiments are reported.

9. Fig 5B, zoomed in insets showing the IBCs in the H&E-stained sections would be helpful. The legend for this figure indicates that “white arrows” point to the IBCs, but these are black in the H&E images.

10. In previous work by Dr. Gilbert (doi.org/10.3389/fcimb.2021.788229), it was reported that deletion of Nur77 had no effect on bacteriuria in a mouse UTI model at 24 hpi. However, in the present study the authors show in Fig 1B that mice lacking Nur77 have reduced levels of bacteriuria at this time point. This may have no real impact on the authors’ overall conclusions, but this discrepancy should probably be acknowledged. Can the authors explain the difference?

Reviewer #2 (Remarks to the Author):

In this manuscript, the authors discovered that Nur77 deficiency promotes the formation of UPEC IBCs in urothelial cells, resulting in persistent infection in bladder tissue. A Nur77 agonist Csn-B was found to inhibit UPEC IBC formation and bladder infection in mice. Although the finding regarding Nur77's potential involvement in UTI is interesting, many of their conclusions lack sufficient support from the provided data. Additionally, the underlying mechanism by which Nur77 affects UPEC IBC formation remains unclear; it is uncertain whether Nur77 directly regulates this process within urothelial cells or through modulation of anti-infection immunity. Therefore, it can be argued that this study does not meet the rigorous criteria set by the journal. Specific points:

1. Considering Nur77's significant regulatory role in immunity, it is important for the authors to investigate whether its regulation of UPEC IBC formation occurs via modulation of anti-infection immunity. The unaltered immune cell count does not exclude this possibility. The interaction between Nur77 and LPS as well as its regulation of NLRP3 further suggest a role in modulating anti-infection immunity. It should also be explored if NLRP3 is involved in regulating UPEC IBC.
2. If Nur77 directly inhibits UPEC IBCs within urothelial cells, what is the underlying mechanism?
3. Figure 4 should include additional indicators to demonstrate urothelial restoration.
4. The experiments presented in Figure 6 should also be conducted using control and Nur77 KO cells.
5. In figure 6E, Csn-B treatment resulted in noticeable urothelial cell death; therefore, it raises concerns about potential bladder tissue damage induced by Csn-B treatment in vivo.

Reviewer #3 (Remarks to the Author):

The manuscript by Collins et al includes analysis of the role of Nur77 in the invasion of uropathogenic *Escherichia coli* (UPEC) into bladder epithelial cells. Invasion leads to the formation of intracellular communities that are recalcitrant to antibiotic treatment. The rise in antibiotic resistant organisms and the inability of most antibiotics to eradicate intracellular communities underscores the critical need for new approaches to treat and prevent this highly prevalent infection. Thus, this study is topical and significant. Based upon their prior work, the authors perform a secondary data analysis to determine that Nur77 is produced in appropriate tissues in the urinary tract. The authors provide complementary in vitro and in vivo experiments using both genetic knockout mice as well as pharmacological inhibitors to determine the role of Nur77 in the pathogenesis of UTIs caused by UPEC. They pay particular attention to the intracellular populations that are resistant to treatment and host eradication. The authors appropriately consider the potential effects of mouse handling in the outcome measures evaluated. The manuscript is interesting, but additional information is needed to ensure appropriate interpretation of some of the data. In particular, there are some questions and concerns regarding how some of the methods are reported.

Minor concerns:

1. Please avoid using terms such as "they" and "it" that personify inanimate objects and is grammatically incorrect.
2. Line 77, the use of "Conversely" at the beginning of the sentence seems confusing as the two complementary approaches indicate the same role for Nur77 and the use of "conversely" suggests the

opposite.

3. Any sentence that refers to prior publications must be cited.

4. The effects of Nur77 on epithelial regeneration is interesting, can you please provide details for how the background levels of staining were distinguished from the CK20 signal?

5. The increase in the number of IBCs in the absence of Nur77 is interesting. However, the statistical differences at 24hours may be driven by the one mouse with over 400 IBCs.

6. The use of the CsnB in the Nur77 knock out strain is an interesting and important control. However, one third of the mice have a higher burden in the bladder can the authors please speculate?

7. The authors provide an appropriate list of limitations of their study, but additional ones remain. The authors compare the knockout to C57/Bl6 and multiple groups have demonstrated that the mouse genetics can influence the outcome of UTI. Jackson labs recommends an F2 hybrid to control for the genetic background of this hybrid strain. In addition, other studies indicate that glucose levels can be modified by CsnB as well as in the Nur77 knock out. Glucose is an important risk factor for UTI.

Major concerns:

1. There needs to be an introduction of CsnB somewhere in the manuscript before the agent is used, without this information, there appears to be no scientific justification fo the use. There is no information regarding the source of CsnB in the methods.

2. For the in vitro studies with CsnB, the authors appropriately monitor the cytotoxicity of the inhibitor. However, the statistically significant effects are observed at 6 hours when the cytotoxicity is observed with CsnB. This is particularly concerning, given that prior studies have indicated cell death of 5637 cells at 6 hours following introduction of UPEC (PMID: 15972487). Potential synergistic effects of the inhibitor and the infection complicate interpretation of the results. Trimethoprim has some ability to cross the membrane, please provide additional information as to the choice of this antibiotic. Additional controls for viability are recommended, or reduction of the data to exclude the 6 hour time point.

3. How was this dosing of CsnB regimen selected? The dosing regimen and the cited paper was one dose and administered in saline not DMSO, please justify changes. Is there anything known about the bioavailability in the tissues and the urine? What is the pharmacological half life, were multiple injections needed?

Dear Editor and Reviewers:

We are submitting our revised manuscript "Nur77 protects the bladder urothelium from intracellular bacterial infection." for your consideration. We have addressed the concerns raised by each reviewer, which has resulted in substantial revision of our manuscript, including the addition of new main and supplementary figures. We have included point-by-point responses to all reviewers' concerns.

Reviewer #1 (Remarks to the Author):

This study by Gilbert and colleagues examines the effects of the nuclear receptor Nur77 on UTI using knockout mice and a Nur77 agonist coupled with bladder cell culture and mouse infection models. This study is among the first to consider the effects of Nur77 on bacterial infection, which is especially interesting given a recent 2023 report indicating that Nur77 can act as a sensor for cytosolic LPS. A previous 2021 study by the corresponding author laid the groundwork for this current submission, presenting evidence that the deletion of Nur77 protects mice from *Gardenerella*-induced resurgence of UPEC reservoirs. Data presented in this submission indicates that Nur77 can affect UPEC titers within the bladder tissue at later time points beyond 2 wpi, and that more IBCs develop with Nur77 KO mice at early time points. In contrast, the Nur77 agonist CsnB is reported to inhibit UPEC invasion of bladder cells as well IBC numbers, levels of bacteriuria, and bacterial titers associated with the bladder tissue. The authors also show that Nur77 deletion does not have any gross effects on immune cell populations within the bladder during UTI, and provide data suggesting that the absence of Nur77 impairs full regeneration of the urothelium at 4 wpi. Overall, the data indicating that Nur77 impacts the longer-term survival of UPEC within the bladder are convincing. However, there are some discrepancies in the data, as noted below, along with some additional controls/suggestions that might help better resolve the effects of Nur77 on UTI and help strengthen the authors' conclusions. The use of CsnB to reduce UPEC titers in the cell culture and mouse models is an especially interesting aspect of this study, but how CsnB is acting to limit UPEC colonization in these assays could be better delineated.

Specific concerns/suggestions:

Major issues

1. Considering results in Fig 5A showing that mice lacking Nur77 have more IBCs at 6 and 24 hpi, it seems surprising that the KO mice do not have higher bladder titers at earlier time points before 3 wpi (Figs 1D and E). Quantifying all intracellular bacteria (e.g. via ex vivo gentamicin protection assays) at these time points could be very helpful in understanding the effects of Nur77. This would also complement the data in Fig 6G indicating that the agonist CsnB inhibits UPEC invasion of bladder cells.

We thank the reviewer for their insightful suggestion. We have performed ex vivo gentamicin protection assays 3, 6 and 24 hpi and have included these data as an additional supplemental figure in the manuscript (Supplemental Figure 4). UPEC CFU in both compartments in Nur77-KO mice was lower 3 hpi, but levels reach the same as WT mice by 6 and 24 hpi. Since IBCs have 10^5 cells, bladders with twice as many IBCs (like in Nur77-KO mice 6 and 24 hpi) will not result in a titer difference that is noticeable on a logarithmic scale. Therefore, the phenotype we observed with IBC enumeration would have been missed with gentamicin protection data alone. The increase in IBC numbers we observed means there are more successful invasion events in bladders of Nur77-KO mice. An increase in the number of infectious foci occurring early during infection is likely to translate into more reservoirs of UPEC persisting in the bladder. Finally, differences in infection can be detected at the titer level at later timepoints because persistent intracellular reservoirs are known to each contain only a few bacterial cells. We have modified the text in this results section to include these details.

2. Lines 246-248, the authors suggest that the “decrease in measured intracellular UPEC caused by CsnB was not solely due to Nur77-driven urothelial cell death.” To better prove this point, the authors could treat the cells with both CsnB and an apoptosis inhibitor (or some similar approach to help rule out host cell death as a confounding factor). The argument that CsnB-induced host cell death is not responsible for the decreased levels of invasion (Fig 6G) does not work so well when considering CsnB effects on UPEC expulsion (Fig 6F and H), which is maximal when host cell death levels are also high.

Multiple reviewers were concerned with our conclusion that urothelial cell death was not responsible for decreased invasion and intracellular UPEC. We apologize that our figure and our description of these experiments in our original manuscript led to misunderstandings about the time points at which these phenotypes were observed in relation to each other. We have attempted to clarify both the figure and the text to better justify our conclusion. Additionally, our original graph showing the % cytotoxicity had tall bars for the 6 h time point which made it appear that there was substantial cytotoxicity. However, the y-axis only went up to 15%. To increase clarity, we now present graphs with the % viability with y-axis going to 100% (now Supplemental Figure 9). We emphasize in the text that the viability of 5637 cells was > 89% across all conditions tested. It is unlikely that such minimal cell death can explain the complete loss of intracellular infection achieved by the highest CsnB dose. Finally, our additional mechanistic insight that CsnB blocks endocytosis (see #) best explains the decrease in UPEC invasion and intracellular infection.

Although our data indicate that urothelial cell death cannot explain the decrease in intracellular UPEC seen 2 and 4 hpi (since cell death was not detectable until 6 hpi), the reviewer’s suggestion to treat cells with both CsnB and an apoptosis inhibitor is a good idea and could still help determine whether execution of the apoptotic pathway is necessary to decrease intracellular UPEC. We performed additional experiments showing that the pan-caspase inhibitor Z-VAD-FMK does not affect the ability of CsnB to block intracellular UPEC infection (Supplemental Figure 9).

2. (cont.) The conclusion that CsnB/Nur77 activation enhances expulsion should be better justified.

We have used the standard assay in the field for measuring UPEC expulsion from urothelial cells. The results show an increase in the percentage of invading UPEC that are expelled (and thus detectable in the culture media) following CsnB treatment. This phenotype was true in both our concurrent treatment and post-invasion treatment *in vitro* models. We believe that the conclusion that CsnB increases expulsion is justified.

2. (cont.) Does CsnB have a general inhibitory effect in endocytotic pathways. For example, will CsnB inhibit uptake of fluorescently-labeled transferrin by 5637 cells?

This is a good question, and our thoughts also went to endocytosis as a potential mechanism. However, we did not originally pursue this hypothesis because our literature searches uncovered no existing evidence that Nur77 or CsnB affect endocytic pathways. (“cytosporone b endocytosis” yields 0 hits on PubMed). Previous studies have shown that UPEC invasion can be blocked by endocytosis inhibitors (dynasore inhibits dynamin activity, EIPA blocks macropinocytosis) {Eto, 2008}. To directly address the reviewers question, we performed endocytosis assays with fluorescently-labeled transferrin with and without CsnB treatment. Surprising to us (given the lack of prior evidence connecting CsnB to endocytosis) we observed that CsnB inhibited transferrin uptake by 5637 cells (Figure 6).

3. Considering the ability of CsnB to induce death of 5637 cells, the authors should determine if the agonist is also inducing death of urothelial cells in the mouse experiments presented in Fig 7.

As described above, and more clearly in our revised manuscript, CsnB induced very minimal death of 5637 cells. However, since the effects of CsnB on the bladder have not been previously investigated, the suggestion to examine the urothelium in mouse experiments is valid. We performed H&E staining

of bladder sections from mice at relevant timepoints in our treatment model. Importantly, this histological analysis did not detect bladder tissue damage or urothelial disruption in bladders from mice treated with CsnB. The superficial cells lining the bladder lumen appear healthy and we saw no evidence of death and exfoliation induced by CsnB.

3. (cont.) Does CsnB inhibit UPEC invasion *in vivo*, as it does in the cell culture assays?

Our data indicate that CsnB inhibits UPEC invasion *in vivo*. We have modified our text to make it clearer that the IBC enumeration is an indicator of UPEC invasion. IBCs represent foci of successful invasion events. CsnB treatment resulted in fewer IBCs *in vivo*. We acknowledge that fewer IBCs being detectable in the bladder would also be true if intracellular replication were inhibited. This is where our *in vitro* experiments help shape our interpretation of the data. Since the results from our *in vitro* infection experiments clearly demonstrate an effect of CsnB on invasion, we conclude that this is the most likely explanation for the difference in IBC numbers *in vivo*. Another consideration is the observation that Nur77-KO mice have increased numbers of IBCs. If Nur77 (and thus CsnB treatment) were modulating intracellular replication and not invasion, we would expect the Nur77-KO mice to have larger IBCs, not necessarily more IBCs. There was no notable difference in IBC size based on our fluorescent microscopy.

Other issues

4. The section starting on page 6 that uses publicly available transcriptomics data to assess Nur77 expression in human and mouse bladder tissues could be presented more concisely, and Fig 2 could be supplemental. For this section, what is the difference between “urothelial cell” and “bladder cell”? Independent confirmation by the authors that Nur77 is expressed in the urothelial cells, and especially the superficial cells, would strengthen this study.

We have trimmed this section to make it more concise and have moved Fig 2 to supplementary material. As to the request for independent confirmation of Nur77 in the bladder by our group, we have included in the figure the Nur77 expression levels in our prior bulk RNA seq data from whole bladder homogenates. These confirm Nur77 expression in the bladders of naïve mice, in mice with UPEC quiescent reservoirs, and mice exposed intravesically to *Gardnerella*. The data from Tabula Muris and Yu et al. demonstrate Nur77 expression in urothelial cells. Yu et al. confirms expression in superficial umbrella cells. The two published scRNAseq papers used slightly different nomenclatures and Yu et al. sub-divided the cell populations at a more granular level than Tabula Muris. In both studies, “urothelial cell” refers to the epithelial cell populations lining the bladder lumen and down to the basement membrane. In Tabula Muris “bladder cell” is a catch-all term for all of the cells that are not leukocytes, endothelial cells, or urothelial cells. Yu et al. had a more in-depth clustering analysis to define these cells, which include fibroblasts, smooth muscle cells, and neurons. Nur77 expression has been reported previously in bladder tumor biopsies and in bladder cancer cell lines

5. Reference to “surface” CK20 as a marker of terminally differentiated umbrella cells (line 197, Fig4B), is a bit confusing, as I don’t think this protein is surface-exposed like the uroplakins. “luminal” might be a better descriptive term. Reduced CK20 staining is the primary/only evidence given to support the idea that regeneration of the urothelium is impaired in Nur77 KO mice. Zoomed in images/insets of some of the umbrella cells depicted in Fig 4A would help the reader better appreciate this phenotype. Use of additional differentiation markers would help back up the conclusions from this section. Is expression of the uroplakins, or their assembly into plaques, diminished in the Nur77 KO mice at 4 wpi? Are size or multi-nucleate status of the umbrella cells altered in the KO? Is regeneration also impaired following injury of the urothelium in the absence of infection (e.g. following cyclophosphamide injury)?

We appreciate the reviewer's interest and suggestions. We recognize that more work is needed to fully decipher the role of Nur77 in urothelial renewal. Since the urothelial renewal phenotype was not a primary focus of our original manuscript, we have decided to remove these data in order to stay focused on the UPEC invasion phenotype for which we have more mechanistic insights.

6. Lines 264 and 334, 5637 cells do not produce uroplakin plaques and do not have fusiform vesicles. The term "endosomes" is more appropriate in this system.

We have changed the text as suggested. Thank you.

7. Why are dotted lines around the different groups of figures included in Fig. 6?

The dotted lines separated the data from "concurrent treatment" and "post-invasion treatment" experiments. We have since reconfigured the figure to make the experimental timelines clearer.

8. Caption for Fig.1D indicates that the data are from 1 experiment. This is not standard in the field, is in contrast to what is stated in the Methods, and different from the other mouse experiments where results from at least two independent experiments are reported.

The data from Fig 1 show no difference between WT and KO mice at either timepoint. It is true that these were from a single experiment. However, we have added data from gentamicin-protection assays (Supplemental Fig. 4; two independent experiments at each time point) that also show that UPEC titers are not different at these time points. Since together these data corroborate the conclusion that UPEC titers are not different, we cannot justify further sacrificing of animals for bulk bladder titers.

9. Fig 5B, zoomed in insets showing the IBCs in the H&E-stained sections would be helpful. The legend for this figure indicates that "white arrows" point to the IBCs, but these are black in the H&E images.

We have added zoomed insets of the IBCs and fixed the legend to correctly indicate arrow colors. Thank you for catching this.

10. In previous work by Dr. Gilbert (doi.org/10.3389/fcimb.2021.788229), it was reported that deletion of Nur77 had no effect on bacteriuria in a mouse UTI model at 24 hpi. However, in the present study the authors show in Fig 1B that mice lacking Nur77 have reduced levels of bacteriuria at this time point. This may have no real impact on the authors' overall conclusions, but this discrepancy should probably be acknowledged. Can the authors explain the difference?

The discrepancy between our prior report and the data presented in this manuscript is most likely due to the inherent variability in UPEC bacteriuria. As can be seen in Figures 1 and 7, there is substantially more spread in UPEC cfu values in urine compared to bladder tissue. This is generally true in published UTI models by our group and others. Greater variability can be expected in urine samples since they are affected by factors like voiding frequency or bladder fullness at sample collection. This is why we do not put an emphasis on the 24 hpi bacteriuria phenotype and, as the reviewer suggested, it has no real impact on our overall conclusions.

Reviewer #2 (Remarks to the Author):

In this manuscript, the authors discovered that Nur77 deficiency promotes the formation of UPEC IBCs in urothelial cells, resulting in persistent infection in bladder tissue. A Nur77 agonist Csn-B was found to inhibit UPEC IBC formation and bladder infection in mice. Although the finding regarding Nur77's potential involvement in UTI is interesting, many of their conclusions lack sufficient support from the provided data. Additionally, the underlying mechanism by which Nur77 affects UPEC IBC formation remains unclear; it is uncertain whether Nur77 directly regulates this process within urothelial cells or through modulation of anti-infection immunity. Therefore, it can be argued that this study does not meet the rigorous criteria set by the journal. Specific points:

1. Considering Nur77's significant regulatory role in immunity, it is important for the authors to investigate whether its regulation of UPEC IBC formation occurs via modulation of anti-infection immunity. The unaltered immune cell count does not exclude this possibility. The interaction between Nur77 and LPS as well as its regulation of NLRP3 further suggest a role in modulating anti-infection immunity. It should also be explored if NLRP3 is involved in regulating UPEC IBC.

We appreciate the reviewer's interest. Since the primary phenotype we observe *in vivo* is the formation of more IBCs in urothelial cells in Nur77-KO mice, we focused the present studies on the urothelium. We agree, and acknowledged in the Discussion, that the unaltered immune cell count does not exclude the possibility of immune mechanisms being involved in how Nur77 modulates IBCs. Deciphering the ways Nur77 might modulate "anti-infection immunity" (which is a quite broad umbrella) during UTI is certainly of interest but beyond the scope of this manuscript.

In response to the reviewer's interest in NLRP3, we will focus on what is known about NLRP3 in infected urothelial cells. A previous study of experimental UTI with UPEC strain CFT073 reported exacerbated inflammation and increased bladder bacterial burden 7 days post infection in NLRP3-deficient mice {Ambite, 2016}. This study did not assess the potential involvement of NLRP3 in regulating UPEC IBC formation. Published *in vitro* studies (e.g., Lindblad et al., 2022) have linked NLRP3 to the proinflammatory responses in urothelial cells that are triggered by infection with UPEC strain CFT073. There are data from *in vitro* infections of urothelial cells (similar to those we performed). Demirel et al. (2018) found that NLRP3-deficient cultured 5637 bladder epithelial cells displayed reduced type 1 fimbriae-mediated invasion by CFT073. The mechanism by which NLRP3 was involved in bacterial invasion of bladder epithelial cells was not explored further and remains unknown. Importantly, this reported observation (decreased invasion in the absence of NLRP3) suggests that activation of NLRP3 in urothelial cells would have the opposite effect (favoring increased invasion). Thus, these published data do not clearly implicate NLRP3 in our findings; if the effect of CsnB in our model was via Nur77-mediated NLRP3 activation, then on the basis of Demirel's findings, we might have expected increased invasion. Since the present data do not indicate involvement of NLRP3 in our observed phenotypes, we have not pursued further investigation of NLRP3 for this manuscript.

2. If Nur77 directly inhibits UPEC IBCs within urothelial cells, what is the underlying mechanism?

We appreciate the reviewer's interest. We do not conclude that Nur77 directly (i.e. via a direct antimicrobial effect) inhibits UPEC within IBCs. The results from Nur77-KO mice and from CsnB-treated mice support the conclusion that Nur77 is involved in limiting UPEC invasion of urothelial cells. We have modified our text to make it clearer that the IBC enumeration is an indicator of UPEC invasion (i.e. that IBCs represent successful invasion events). Nur77-KO mice harbored more IBCs 6 hpi and higher bladder reservoir titers 3 and 4 wpi, both of which require UPEC invasion of urothelial cells. CsnB treatment resulted in fewer IBCs *in vivo* and blocked UPEC invasion *in vitro*. We have added data demonstrating that CsnB prevents endocytosis in urothelial cells *in vitro*. These findings

suggest that Nur77 participates in endocytosis or vesicle trafficking, which is a novel function for Nur77. As such, substantial further studies outside the scope and focus of this manuscript would be required to determine the underlying mechanism by which Nur77 might regulate endocytosis or vesicle trafficking.

3. Figure 4 should include additional indicators to demonstrate urothelial restoration.

We recognize that more work is needed to fully decipher the role of Nur77 in urothelial renewal. Reviewer #1 had the same concern. Since the urothelial renewal phenotype was not a primary focus of our original manuscript, we have decided to remove these data to stay more focused on the UPEC invasion phenotype for which now we have more mechanistic insights.

4. The experiments presented in Figure 6 should also be conducted using control and Nur77 KO cells.

We performed siRNA on 5637 cells and observed no difference in invasion or intracellular infection (added as Supplemental Figure X). Therefore, although activation of Nur77 is sufficient to limit infection of 5637 cells, Nur77 is not essential to permit invasion and intracellular infection *in vitro*.

5. In figure 6E, Csn-B treatment resulted in noticeable urothelial cell death; therefore, it raises concerns about potential bladder tissue damage induced by Csn-B treatment *in vivo*.

We recognize that our presentation of the data made our interpretation of the results unclear. Other reviewers were also concerned with cell death. It is true that cell death was noticeable *in vitro*, but the levels were very low, with the highest dose of CsnB causing only 11% cytotoxicity. Our original graph of the data as % cytotoxicity, with the y-axis scale only going to 15 made it appear that cell death levels were high. We now present the same data as % viability to more clearly show that CsnB caused minimal cell death. Cell viability was >89% for all conditions tested. Our additional data from cells co-treated with CsnB and the caspase inhibitor Z-VAD-FMK provide additional evidence that cell death is not required for CsnB to inhibit UPEC infection.

The reviewer's concern about the effect of CsnB *in vivo* is still a valid one (also shared by Reviewer 1) since the effects of CsnB on the bladder have not been previously investigated. We performed H&E staining of bladder sections from mice at relevant timepoints in our treatment model. Importantly, this histological analysis did not detect bladder tissue damage or urothelial disruption in bladders from mice treated with CsnB. The superficial cells lining the bladder lumen appear healthy and we saw no evidence of death and exfoliation induced by CsnB.

Reviewer #3 (Remarks to the Author):

The manuscript by Collins et al includes analysis of the role of Nur77 in the invasion of uropathogenic *Escherichia coli* (UPEC) into bladder epithelial cells. Invasion leads to the formation of intracellular communities that are recalcitrant to antibiotic treatment. The rise in antibiotic resistant organisms and the inability of most antibiotics to eradicate intracellular communities underscores the critical need for new approaches to treat and prevent this highly prevalent infection. Thus, this study is topical and significant. Based upon their prior work, the authors perform a secondary data analysis to determine that Nur77 is produced in appropriate tissues in the urinary tract. The authors provide complementary *in vitro* and *in vivo* experiments using both genetic knockout mice as well as pharmacological inhibitors to determine the role of Nur77 in the pathogenesis of UTIs caused by UPEC. They pay particular attention to the intracellular populations that are resistant to treatment and host eradication. The authors appropriately consider the potential effects of mouse handling in the outcome measures evaluated. The manuscript is interesting, but additional information is needed to ensure appropriate interpretation of some of the data. In particular, there are some questions and concerns regarding how some of the methods are reported.

Minor concerns:

1. Please avoid using terms such as “they” and “it” that personify inanimate objects and is grammatically incorrect.

We have fixed this in our manuscript.

2. Line 77, the use of “Conversely” at the beginning of the sentence seems confusing as the two complementary approaches indicate the same role for Nur77 and the use of “conversely” suggests the opposite.

We intended “conversely” to refer to the phenotypes themselves being opposite (higher UPEC infection vs. lower UPEC infection), but we recognize the potential confusion. We revised these lines to remove “conversely” and instead refer to “complimentary findings.”

3. Any sentence that refers to prior publications must be cited.

We found that citations to our prior publications in the “Study Design” section of materials and methods were missing and we have added them to our revision. Other references to prior publications are cited throughout the manuscript.

4. The effects of Nur77 on epithelial regeneration is interesting, can you please provide details for how the background levels of staining were distinguished from the CK20 signal?

In recognition of the concern raised by other reviewers that more data are needed to define the epithelial regeneration phenotype, we have removed the CK20 data from this manuscript.

5. The increase in the number of IBCs in the absence of Nur77 is interesting. However, the statistical differences at 24hours may be driven by the one mouse with over 400 IBCs.

Thank you for bringing this up. We have confirmed that the difference between WT and Nur77-KO mice remains statistically significant even if we were to eliminate the >400 data point.

6. The use of the CsnB in the Nur77 knock out strain is an interesting and important control. However, one third of the mice have a higher burden in the bladder can the authors please speculate?

It is typical for one or two mice (including WT mice) in a given experiment to have higher bacterial burden at later time points, including 1 wpi (see Fig 1D and E). Prior studies have found that ~10-30% of C57Bl/6 mice will display persistent bacteriuria and higher bladder burden following experimental UTI. Our data are in line with the UTI mouse model field.

7. The authors provide an appropriate list of limitations of their study, but additional ones remain. The authors compare the knockout to C57/Bl6 and multiple groups have demonstrated that the mouse genetics can influence the outcome of UTI. Jackson labs recommends an F2 hybrid to control for the genetic background of this hybrid strain. In addition, other studies indicate that glucose levels can be modified by CsnB as well as in the Nur77 knock out. Glucose is an important risk factor for UTI.

Thank you for bringing up these points. We have added the following acknowledgment to our manuscript: “We compared Nur77-KO mice to wild type C57BL/6J to be consistent with the majority of published studies that have used the Nur77-KO strain. An alternative option for future studies would be to use the B6129PF2/J that is an F2 hybrid from C57BL/6J females (B6) and 129P3/J males (129P) approximate controls for genetically engineered strains that were generated with 129-derived embryonic stem cells, which is how the Nur77-KO strain was generated.”

As to the question of glucose levels playing a role in our observed phenotypes, we do not think this is likely. The reviewer is correct that a high dose of CsnB (50 mg/kg) has been shown to increase blood glucose levels 30 min after i.p. injection, but this effect wanes by 2 hours. Mice with high blood glucose (streptozocin-induced diabetic model) have increased susceptibility to UTI, with higher bacterial burdens 6, 24 and 72 hpi. Since mice treated with CsnB had decreased susceptibility to UTI, it is unlikely that glucose levels are playing a substantial role in our model. We have added this discussion to our manuscript.

Major concerns:

1. There needs to be an introduction of CsnB somewhere in the manuscript before the agent is used, without this information, there appears to be no scientific justification for the use. There is no information regarding the source of CsnB in the methods.

We moved our description of CsnB from the discussion section to the results section in which we performed experiments with CsnB. We also added text to justify why we used CsnB. We added the manufacturer and product number to our materials and methods section.

“The increase in IBC numbers in Nur77-KO mice suggested that Nur77 could function in urothelial cells to limit intracellular UPEC burden. To test this idea, we used the Nur77 agonist cytosporone B (CsnB) (43) to activate Nur77 in cultured 5637 urothelial cells. CsnB, is an octaketide isolated from an endophytic fungus that was the first identified natural ligand for Nur77 (43). CsnB has a strong affinity for Nur77 and has been widely used to specifically activate Nur77 *in vitro* {cite}. Therefore, CsnB would allow us to determine the effect of activating Nur77 at various timepoints during UPEC infection. Given that Nur77-KO mice had increased UPEC infection, we hypothesized that activation of Nur77 with CsnB would decrease UPEC urothelial cell infection.”

2. For the *in vitro* studies with CsnB, the authors appropriately monitor the cytotoxicity of the inhibitor. However, the statistically significant effects are observed at 6 hours when the cytotoxicity is observed with CsnB. This is particularly concerning, given that prior studies have indicated cell death of 5637 cells at 6 hours following introduction of UPEC (PMID: 15972487). Potential synergistic effects of the inhibitor and the infection complicate interpretation of the results. Trimethoprim has some ability to cross the membrane, please provide additional information as to the choice of this antibiotic. Additional controls for viability are recommended, or reduction of the data to exclude the 6 hour time point.

The other reviewers also had concerns about cytotoxicity. We think this is due to lack of clarity in how we presented the timeline of our experiments in relation to the data. Cell death was minimal and did not occur until 4 hpi, whereas UPEC intracellular infection differences were apparent by 30 min. We have added additional experiments with a caspase inhibitor that are relevant to this concern. Please refer to our responses to Reviewer 1, comment #2 and Reviewer 2, comment #5 for additional details.

The concern about UPEC-induced cell death is valid (although, the PMID referenced by the reviewer is a paper that does not have any cell death data in it). To address potential synergistic effects of the inhibitor, we have included additional data from cells infected with UPEC and treated with CsnB (**Fig. S9**). These also show very minimal effects on cell viability. The reviewer suggested excluding the 6 h time point. Since many readers are likely to be familiar with Nur77's role in apoptosis, they are likely to wonder whether CsnB can trigger death of urothelial cells. Therefore, we have still included the data from all timepoints in our supplemental figure (**Fig S9**).

As to using trimethoprim, we followed the protocol for expulsion that have been established by the field {Miao, 2015; Miao 2017}. Trimethoprim did not appear to cross into cells and kill intracellular bacteria in our assay because we still detected intracellular UPEC cfu in urothelial cells treated with trimethoprim at similar levels as we typically see in experiments with gentamicin.

3. How was this dosing of CsnB regimen selected? The dosing regimen and the cited paper was one dose and administered in saline not DMSO, please justify changes. Is there anything known about the bioavailability in the tissues and the urine? What is the pharmacological half life, were multiple injections needed?

We apologize that our description was not clear. We also administered CsnB in saline. For preparation of CsnB injections, we diluted a CsnB stock solution (reconstituted in DMSO) in saline. For the vehicle control, we diluted an equal volume of DMSO in saline. There is not a consensus in the literature regarding dosing regimens for CsnB, but most studies we found administered the drug multiple times. The study we originally cited described administering 5 mg/kg CsnB i.p. daily over the course of their 7 day or 16 day influenza virus infection model. Ding et al. {2022} administered 10 mg/kg i.p. CsnB every other day in their study of acute cardiac allograft rejection. We are not aware of a study reporting the pharmacological half-life of CsnB in mice.

REVIEWERS' COMMENTS

Reviewer #1 (Remarks to the Author):

The authors of this revised paper addressed all of my previous concerns. This study will be a valuable contribution to the UTI field and related areas of research. Nice work!

Reviewer #2 (Remarks to the Author):

Although the authors have partially addressed my concerns, I believe that not all of the concerns raised by me and other reviewers have been adequately resolved. Simply removing data without addressing the concerns is not a constructive approach; it would be more beneficial to provide further clarification by experimental evidence.

My primary concern pertains to the lack of mechanism analysis. While it is intriguing that CsnB may restrict intracellular UPEC infection through endocytosis inhibition, whether Nur77 plays a role in regulating endocytosis remains unknown. The mechanism underlying CsnB-induced expulsion of UPEC also lacks clarity. Additionally, the authors claim that “although activation of Nur77 is sufficient to limit infection of 5637 cells, Nur77 is not essential to permit invasion and intracellular infection in vitro “. This statement lacks persuasiveness since differences in IBCs could be observed between WT and KO mice which were not administered with CsnB.

Reviewer #3 (Remarks to the Author):

The Authors have been very comprehensive, considerate and respectful in their completion of additional experiments and adjusting the manuscript to appropriately address the reviewers concerns. I have no additional concerns.